# Randomized algorithms and PAC bounds for inverse reinforcement learning in continuous spaces

**Angeliki Kamoutsi**
EPFL, Switzerland
angeliki.kamoutsi@epfl.ch

**Peter Schmitt-Förster**
University of Konstanz, Germany
peter.schmitt-foerster@uni-konstanz.de

**Tobias Sutter**
University of Konstanz, Germany
tob.sutter@uni-konstanz.de

**Volkan Cevher**
EPFL, Switzerland
volkan.cevher@epfl.ch

**John Lygeros**
ETH Zürich, Switzerland
jlygeros@ethz.ch

## Abstract

This work studies discrete-time discounted Markov decision processes with continuous state and action spaces and addresses the inverse problem of inferring a cost function from observed optimal behavior. We first consider the case in which we have access to the entire expert policy and characterize the set of solutions to the inverse problem by using occupation measures, linear duality, and complementary slackness conditions. To avoid trivial solutions and ill-posedness, we introduce a natural linear normalization constraint. This results in an infinite-dimensional linear feasibility problem, prompting a thorough analysis of its properties. Next, we use linear function approximators and adopt a randomized approach, namely the scenario approach and related probabilistic feasibility guarantees, to derive $\varepsilon$-optimal solutions for the inverse problem. We further discuss the sample complexity for a desired approximation accuracy. Finally, we deal with the more realistic case where we only have access to a finite set of expert demonstrations and a generative model and provide bounds on the error made when working with samples.

## 1 Introduction

In the standard reinforcement learning (RL) setting [1, 2, 3, 4], a cost signal is given to instruct agents on completing a desired task. However, oftentimes, it is either too challenging to optimize a given cost (e.g., due to sparsity), or it is prohibitively hard to manually engineer a cost function that induces complex and multi-faceted optimal behaviors. At the same time, in many real-world scenarios, encoding preferences using expert demonstrations is easy and provides an intuitive and human-centric interface for behavioral specification [5, 6, 7]. Considering the inverse reinforcement (IRL) problem involves deducing a cost function from observed optimal behavior. IRL is actively researched with applications in engineering, operations research, and biology [8, 9, 10]. There are two main motivations behind inverse decision-making. The first one concerns situations where the cost function is of interest by itself, e.g., for scientific inquiry, modeling of human and animal behavior [11, 12] or modeling of other cooperative or adversarial agents [13]. The second one concerns the task of imitation or apprenticeship learning [14] by first recovering the expert's cost function and then using it to reproduce and synthesize the optimal behavior. For instance, in engineering, IRL can be used to explain and imitate the observed expert behavior, e.g., in the highway driving task [14, 15], parking lot navigation [16], and urban navigation [17]. Other examples can be found in humanoid robotics and understanding of human locomotion [18]. Despite extensive research efforts, our understanding of IRL still has significant limitations. One major gap lies in the absence of algorithms designed for continuous state and action spaces, which are crucial for numerous promising applications like

autonomous vehicles and robotics that operate in continuous environments. Most existing state-of-the-art IRL algorithms for the continuous setting often adopt a policy-matching approach instead of directly solving the IRL problem [19, 20, 21, 22, 23, 24, 25, 26, 27]. However, this approach tends to provide a less robust representation of agent preferences [26], since the recovered policy is highly dependent on the environment dynamics. State-of-the-art IRL algorithms are empirically successful but lack formal guarantees. Theoretical assurances are crucial for practical implementation, especially in safety-critical systems with potential fatal consequences.

**Contributions.**   This work deals with discrete-time Markov decision processes (MDPs) on continuous state and action spaces under the total expected discounted cost optimality criterion and studies the inverse problem of inferring a cost function from observed optimal behavior. Under the assumption that the control model is Lipschitz continuous, we propose an optimization-based framework to infer the cost function of an MDP given a generative model and traces of an optimal policy. Our approach is based on the linear programming (LP) approach to continuous MDPs [28], complementary slackness optimality conditions, related developments in randomized convex optimization [29, 30, 31] and uniform finite sample bounds from statistical learning theory [32].

To this aim, we first consider the case in which we have access to the entire optimal policy $\pi_E$ and starting from the LP formulation of the MDP, we characterize the set of solutions to the inverse problem by using occupation measures, linear duality and complementary slackness conditions. This results in an infinite-dimensional linear feasibility problem. Although from a theoretical point of view our approach succeeds in characterizing inverse optimality in its full generality, in practice the following important challenges need to be addressed. First, the inverse problem is ill-conditioned and ill-posed, since each task is consistent with many cost functions. Thus a main challenge is coming up with a meaningful one. To this end, we enforce an additional natural linear normalization constraint in order to avoid trivial solutions and ill-posedness. Another challenge is the infinite-dimensionality of the LP formulation, which makes it computationally intractable. To alleviate this difficulty, we propose an approximation scheme that involves a restriction of the decision variables from an infinite-dimensional function space to a finite dimensional subspace (tightening), followed by the approximation of the infinitely-uncountably-many constraints by a finite subset (relaxation). In particular, we use linear function approximators and adopt a randomized approach, namely the scenario approach [29, 33], and related probabilistic feasibility guarantees [30], to derive $\varepsilon$-optimal solutions for the inverse problem, as well as explicit sample complexity bounds for a desired approximation accuracy. Finally, we deal with the more realistic case where we only have access to a finite set of expert demonstrations and a generative model and provide bounds on the error made when working with samples.

**Related literature.**   Our principal aim is to address problems with uncountably infinitely many states and actions. Existing IRL algorithms treat the unknown cost function as a linear combination [34, 35, 15, 17, 36, 37] or nonlinear function [38, 39, 40] of features. In particular, there are three broad categories of formulations. In *feature expectation matching* [35, 15, 17] one attempts to match the feature expectation of a policy to the expert, while in *maximum margin planning* [36, 38, 40] the goal is to learn mappings from features to cost functions so that the demonstrated policy is better than any other policy by a margin defined by a loss-function. Moreover, a *probabilistic approach* is to interpret the cost function as parametrization of a policy class such that the true cost function maximizes the likelihood of observing the demonstrations [17, 37, 39, 41]. Most existing IRL methods that recover a cost function, are either designed exclusively for MDPs with finite state and action spaces, or rely on an oracle access to an RL solver which is used repeatedly in the inner loop of an iterative procedure. An exception are the works [42, 43, 44, 22, 25, 45, 26], that perform well in the experimental settings considered, without providing theoretical guarantees. Relying on oracle access to an RL solver is a significant computational burden for applying these methods to MDPs with continuous state and action spaces since solving a continuous MDP is a challenging and computationally expensive problem on its own. As a result, IRL over uncountable spaces remains largely unexplored. In this work we aim to contribute to this line of research and propose a method that avoids repeatedly solving the forward problem and simultaneously provides probabilistic performance guarantees on the quality of the recovered solution.

Linear duality and complementarity were first proposed in [46] for solving finite-dimensional inverse LPs. The idea was then extended to inverse conic optimization problems in [47] by using KKT optimality conditions. The fundamental difference between these works and the present paper is that they deal with finite-dimensional convex optimization programs where the agent has complete knowledge of the optimal behavior as a finite-dimensional vector. In our setting we have the additional

difficulty of the infinite-dimensional and data-driven nature of the problem. In [48] the authors use occupation measures and complementarity in linear programming to formulate the inverse deterministic continuous-time optimal control problem. Under the assumption of polynomial dynamics and semi-algebraic state and input constraints, they propose an approximation scheme based on sum-of-squares semidefinite programming. Contrary to [48], we consider the problem of inverse discrete-time stochastic optimal control. In such a stochastic environment, assuming polynomial dynamics clearly is restrictive, excluding any setting with Gaussian noise, e.g., the LQG problem. Our approach is not limited to the case of polynomial dynamics and semi-algebraic constraints but is able to tackle the general case, while also providing performance guarantees as in [48].

Our work is closely related to the recent theoretical works on IRL [49, 50, 51, 52, 53, 54, 55, 56]. However, these papers consider either tabular MDPs [49, 50, 52, 53, 54, 55] or MDPs with continuous states and finite action spaces [51, 56]. In contrast, our contribution delves into the theoretical analysis of IRL in the intricate landscape of continuous state and action spaces. Notably, our framework, when applied to finite tabular MDPs and a stationary Markov expert policy $\pi_E$, simplifies to the inverse feasibility set considered in [52, 53, 54] (see also Appx A.2). The methodology put forth in those studies, extends the LP formulation previously explored in [12, 49, 50, 51], which primarily dealt with deterministic expert policies of the form $\pi_E \equiv a_1$. In our work, by using occupancy measures instead of policies and employing Lagrangian duality, we are able to characterize inverse feasibility for general continuous MDPs regardless of the complexity of the expert policy. Moreover, our framework empowers us to leverage offline expert demonstrations to compute an approximate feasibility set and recover a cost through a sample-based convex program. This flexibility surpasses previous theoretical IRL settings, where either $\pi_E$ is assumed to be fully known [49, 51, 52] or active querying of $\pi_E$ is possible for each state [53, 54]. Finally, our assumption of a Lipschitz MDP model is milder and more general than the infinite matrix representation considered in [51], thus accommodating a broader range of MDP models. Overall, we establish a link between our methodology and the existing body of literature on LP formulations for IRL, while also accounting for continuous states and action spaces and more general expert policies. Finally, we would like to highlight a key distinction between our work and recent theoretical IRL papers [52, 53, 54, 55]. Unlike these recent works, our study goes beyond the examination of the properties of the inverse feasibility set and its estimated variant. Our contribution extends to tackling the reward ambiguity problem, a well-known limitation of the IRL paradigm, and provides theoretical results in this direction. Additionally, our work introduces function approximation techniques that come with robust theoretical guarantees. Finally, we study how constraint sampling in infinite-dimensional LPs can be exploited to derive a single nearly optimal solution with probabilistic performance guarantees.

**Basic definitions and notations.** Let $(X, \rho)$ be a *Borel space*, i.e., $X$ is a Borel subset of a complete and separable $\rho$-metric space, and let $\mathcal{B}(X)$ be its Borel $\sigma$-algebra. We denote by $\mathcal{M}(X)$ the Banach space of finite signed Borel measures on $X$ equipped with the total variation norm and by $\mathcal{P}(X)$ the convex set of Borel probability measures. Let $\delta_x \in \mathcal{P}(X)$ be the Dirac measure centered at $x \in X$. Measurability is always understood in the sense of Borel measurability. An open ball in $(X, \rho)$ with radius $r$ and center $x_0$ is denoted by $B_r(x_0) = \{x \in X : \rho(x, x_0) < r\}$. Given a measurable function $u : X \to \mathbb{R}$, its sup-norm is given by $\|u\|_\infty \triangleq \sup_{x \in X} |u(x)|$. Moreover, we define the Lipschitz semi-norm by $|u|_L \triangleq \sup_{x \neq x'} \left\{ \frac{|u(x) - u(x')|}{\rho(x, x')} \right\}$ and the Lipschitz norm by $\|u\|_L \triangleq \|u\|_\infty + |u|_L$. Let $\mathrm{Lip}(X)$ be the Banach space of real-valued bounded Lipschitz continuous functions on $X$ together with the Lipschitz norm $\|\cdot\|_L$. Then, $(\mathcal{M}(X), \mathrm{Lip}(X))$ forms a *dual pair* of vector spaces with duality brackets $\langle \mu, u \rangle \triangleq \int_{\mathcal{X}} u(x) \mathrm{d}\mu$, for all $\mu \in \mathcal{M}(X)$, $u \in \mathrm{Lip}(X)$. Moreover, if $\mathcal{M}(X)_+$ is the convex cone of finite nonnegative Borel measures on $X$, then its dual convex cone is the set $\mathrm{Lip}(X)_+$ of nonnegative bounded and Lipschitz continuous functions on $X$. Under the additional assumption that $X$ is compact, the *Wasserstein norm* $\|\cdot\|_W$ on $\mathcal{M}(X)$ is dual to the Lipschitz norm, i.e., $\|\mu\|_W \triangleq \sup_{\|u\|_L \leq 1} \langle \mu, u \rangle$. If $X, Y$ are Borel spaces, a *stochastic kernel* on $X$ given $Y$ is a function $P(\cdot|\cdot) : \mathcal{B}(X) \times Y \to [0, 1]$ such that $P(\cdot|y) \in \mathcal{P}(X)$, for each fixed $y \in Y$, and $P(B|\cdot)$ is a measurable real-valued function on $Y$, for each fixed $B \in \mathcal{B}(X)$.

## 2 Markov decision processes and linear programming formulation

**Continuous Markov decision process.** Consider a *Markov decision process* (MDP) given by a tuple $\mathcal{M}_c \triangleq (\mathcal{X}, \mathcal{A}, \mathsf{P}, \gamma, \nu_0, c)$, where $\mathcal{X}$ is a Borel space called the *state space*, $\mathcal{A}$ is a Borel space

called the *action space*, P is a stochastic kernel on $\mathcal{X}$ given $\mathcal{X} \times \mathcal{A}$ called the *transition law*, $\gamma \in (0,1)$ is the *discount factor*, $\nu_0 \in \mathcal{P}(\mathcal{X})$ is the *initial probability distribution*, and $c : \mathcal{X} \times \mathcal{A} \to \mathbb{R}$ is the *cost function*. The model $\mathcal{M}_c$ represents a controlled discrete-time stochastic system with initial state $x_0 \sim \nu_0(\cdot)$. At time step $t$, if the system is in state $x_t = x \in \mathcal{X}$, and the action $a_t = a \in \mathcal{A}$ is taken, then a corresponding cost $c(x,a)$ is incurred, and the system moves to the next state $x_{t+1} \sim \mathsf{P}(\cdot|x,a)$. Once transition into the new state has occurred, a new action is chosen, and the process is repeated. A *stationary Markov policy* $\pi$ is a stochastic kernel on $\mathcal{A}$ given $\mathcal{X}$ and $\pi(\cdot|x) \in \mathcal{P}(\mathcal{A})$ denotes the probability distribution of the action $a_t$ taken at time $t$, while being in state $x$. We denote the space of stationary Markov policies by $\Pi_0$. Given a policy $\pi$, we denote by $\mathbb{P}_{\nu_0}^\pi$ the induced probability measure[1] on the canonical sample space $\Omega \triangleq (\mathcal{X} \times \mathcal{A})^\infty$, i.e., $\mathbb{P}_{\nu_0}^\pi[\cdot] = \mathrm{Prob}[\cdot \mid \pi, x_0 \sim \nu_0]$ is the probability of an event when following $\pi$ starting from $x_0 \sim \nu_0$. The expectation operator with respect to the trajectories generated by $\pi$ when $x_0 \sim \nu_0$, is denoted by $\mathbb{E}_{\nu_0}^\pi$. If $\nu_0 = \delta_x$ for some $x \in \mathcal{X}$, then we will write for brevity $\mathbb{P}_x^\pi$ and $\mathbb{E}_x^\pi$.

The optimal control problem we are interested in is[2]

$$V_c^\star(\nu_0) \triangleq \min_{\pi \in \Pi_0} V_c^\pi(\nu_0), \qquad\qquad \text{(MDP}_\mathbf{c}\text{)}$$

where $V_c^\pi(\nu_0) \triangleq \mathbb{E}_{\nu_0}^\pi [\sum_{t=0}^\infty \gamma^t c(x_t, a_t)]$. A policy $\pi^\star$ is called $\gamma$-*discounted* $\nu_0$-*optimal* if $V_c^{\pi^\star}(\nu_0) = V_c^\star(\nu_0)$, and the *optimal value function* $V_c^\star : \mathcal{X} \to \mathbb{R}$ is given by $V_c^\star(x) \triangleq V_c^\star(\delta_x)$. We impose the following assumptions on the MDP model which hold throughout the article. These are the usual continuity-compactness conditions [59], together with the Lipschitz continuity of the elements of the MDP; see, e.g., [60]. We recall that the transition law P acts on bounded measurable functions $u : \mathcal{X} \to \mathcal{R}$ from the left as $\mathsf{P}u(x,a) \triangleq \int_\mathcal{X} u(y)\mathsf{P}(\mathrm{d}y|x,a)$, for all $(x,a) \in \mathcal{X} \times \mathcal{A}$.

**Assumption 2.1** (Lipschitz control model)**.**

*(A1)* $\mathcal{X}$ *and* $\mathcal{A}$ *are compact subsets of Euclidean spaces;*

*(A2)* *the transition law* P *is* weakly continuous*, meaning that* P$u$ *is continuous on* $\mathcal{X} \times \mathcal{A}$*, for every continuous function* $u : \mathcal{X} \to \mathbb{R}$*. Moreover,* P *is* Lipschitz continuous*, i.e., there exists a constant* $L_\mathsf{P} > 0$ *such that for all* $(x,a),(y,b) \in \mathcal{X} \times \mathcal{A}$ *and all* $u \in \mathrm{Lip}(\mathcal{X})$*, it holds that* $|\mathsf{P}u(x,a) - \mathsf{P}u(y,b)| \le L_\mathsf{P}|u|_\mathsf{L}(\|x-y\|_2 + \|a-b\|_2)$*;*

*(A3)* *the cost function* $c$ *is in* $\mathrm{Lip}(\mathcal{X} \times \mathcal{A})$ *with Lipschitz constant* $L_c > 0$*. That is, for all* $(x,a),(y,b) \in \mathcal{X} \times \mathcal{A}$*,* $|c(x,a) - c(y,b)| \le L_c(\|x-y\|_2 + \|a-b\|_2)$*;*

Note that Assumption2.1 (A2) is fulfilled when the transition law P has a density function $f(y,x,a)$ that is Lipschitz continuous in $y$ uniformly in $(x,a)$ [60]. This encompasses various probability distributions, such as the uniform, Gaussian, exponential, Beta, Gamma, and Laplace distributions, among others. Additionally, it applies to the infinite matrix representation considered in[51]. Consequently, Assumption 2.1 accommodates a broad range of MDP models and allows for the consideration of smooth and continuous dynamics that reflect the characteristics of several real-world applications, such as robotics, or autonomous driving. Importantly, Assumption 2.1 ensures that the value function $V_c^\star$ is in $\mathrm{Lip}(\mathcal{X})$ and is uniquely characterized by the *Bellman optimality equation* $V_c^\star(x) = \min_{a \in \mathcal{A}}\{c(x,a) + \gamma \int_\mathcal{X} V_c^\star(y)\mathsf{P}(dy|x,a)\}$, for all $x \in \mathcal{X}$ [60, Thm. 3.1] and [61].

**Occupancy measures.** For every policy $\pi$, we define the *occupancy measure* $\mu_{\nu_0}^\pi \in \mathcal{M}(\mathcal{X} \times \mathcal{A})_+$ by $\mu_{\nu_0}^\pi(E) \triangleq \sum_{t=0}^\infty \gamma^t \mathbb{P}_{\nu_0}^\pi[(x_t, a_t) \in E], E \in \mathcal{B}(\mathcal{X} \times \mathcal{A})$. The occupancy measure can be interpreted as the discounted visitation frequency of the set $E$ when acting according to policy $\pi$. The set of occupancy measures is characterized in terms of linear constraint satisfaction [62, Theorem 6.3.7]. To this end consider the convex set of measures, $\mathfrak{F} \triangleq \{\mu \in \mathcal{M}(\mathcal{X} \times \mathcal{A})_+ : T_\gamma\mu = \nu_0\}$, where $T_\gamma : \mathcal{M}(\mathcal{X} \times \mathcal{A}) \to \mathcal{M}(\mathcal{X})$ is a linear and weakly continuous operator given by

$$(T_\gamma\mu)(B) \triangleq \mu(B \times \mathcal{A}) - \gamma \int_{\mathcal{X} \times \mathcal{A}} \mathsf{P}(B|x,a)\mu(\mathrm{d}(x,a)),$$

---

[1] Note that $\mathbb{P}_{\nu_0}^\pi$ is uniquely determined by the transition law P, the initial state distribution $\nu_0$ and the policy $\pi$ [57, Prop. 7.28].

[2] For the discounted policy optimization problem considered in this paper it suffices to restrict our search to stationary Markov policies, [58, Thm. 5.5.3]. However, the expert policy $\pi_\mathsf{E}$ can be nonstationary and history-dependent.

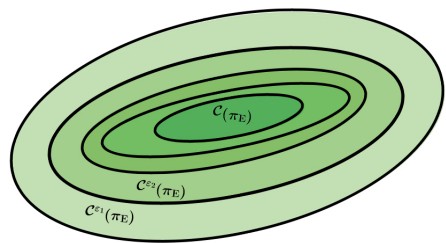

Figure 1: Illustration of Theorem 3.1 for $\varepsilon_1 > \varepsilon_2$.

for all $B \in \mathcal{B}(\mathcal{X})$. Then, $\mathfrak{F} = \{\mu_{\nu_0}^\pi : \pi \in \Pi_0\}$. Moreover, $\langle \mu_{\nu_0}^\pi, c \rangle = V_c^\pi(\nu_0)$, for every $\pi$.

**The linear programming approach.** A direct consequence is that ($\text{MDP}_\mathbf{c}$) can be stated equivalently as an infinite-dimensional LP over measures

$$\mathcal{J}_c(\nu_0) \triangleq \inf_{\mu \in \mathcal{M}(\mathcal{X} \times \mathcal{A})_+} \{\langle \mu, c \rangle : T_\gamma \mu = \nu_0\}. \tag{$\text{P}_\mathbf{c}$}$$

In particular the infimum in ($\text{P}_\mathbf{c}$) is attained and $\pi^\star$ is optimal for ($\text{MDP}_\mathbf{c}$) if and only if $\mu_{\nu_0}^{\pi^\star}$ is optimal for the primal LP ($\text{P}_\mathbf{c}$). The dual LP of ($\text{P}_\mathbf{c}$) is given by

$$\mathcal{J}_c^*(\nu_0) \triangleq \sup_{u \in \text{Lip}(\mathcal{X})} \{\langle \nu_0, u \rangle : c - T_\gamma^* u \geq 0 \text{ on } \mathcal{X} \times \mathcal{A}\}, \tag{$\text{D}_\mathbf{c}$}$$

where the adjoint linear operator $T_\gamma^* : \text{Lip}(\mathcal{X}) \to \text{Lip}(\mathcal{X} \times \mathcal{A})$ of $T_\gamma$ is given by

$$(T_\gamma^* u)(x, a) \triangleq u(x) - \gamma \int_{\mathcal{X}} u(y) \mathsf{P}(\mathrm{d}y | x, a).$$

Under Assumption 2.1, $T_\gamma^*$ is well-defined and the dual LP ($\text{D}_\mathbf{c}$) is solvable, i.e., the supremum is attained, and strong duality holds. That is, $\mathcal{J}_c(\nu_0) = \mathcal{J}_c^*(\nu_0) = V_c^\star(\nu_0)$. In particular, the value function $V_c^\star$ is an optimal solution for the dual LP ($\text{D}_\mathbf{c}$). More details on the LP formulations for MDPs can be found in Appendix A.1.

## 3 Inverse reinforcement learning and characterization of solutions

We first define the *inverse reinforcement learning* (IRL) problem and the *inverse feasibility set*.

**Definition 3.1** (IRL [12, 52])**.** *An IRL problem is a pair $\mathcal{B} \triangleq (\mathcal{M}, \pi_\text{E})$, where $\mathcal{M} \triangleq (\mathcal{X}, \mathcal{A}, \mathsf{P}, \nu_0, \gamma)$ is an MDP without cost function and $\pi_\text{E}$ is an observed expert policy. We say that $c \in \text{Lip}(\mathcal{X} \times \mathcal{A})$ is* inverse feasible *for $\mathcal{B}$, if $\pi_\text{E}$ is a $\gamma$-discount $\nu_0$-optimal policy for ($\text{MDP}_\mathbf{c}$) with cost c. The set of all $c \in \text{Lip}(\mathcal{X} \times \mathcal{A})$ that are inverse feasible is called the* inverse feasibility set *and is denoted by $\mathcal{C}(\pi_\text{E})$.*

Next, we use the primal-dual LP approach to MDPs and complementary slackness to characterize $\mathcal{C}(\pi_\text{E})$. To this end, we first define the $\varepsilon$-*inverse feasibility set* $\mathcal{C}^\varepsilon(\pi_\text{E})$.

**Definition 3.2.** *Let $\varepsilon \geq 0$. We say that a cost function c is $\varepsilon$-inverse feasible for $\mathcal{B} = (\mathcal{M}, \pi_\text{E})$ and denote $c \in \mathcal{C}^\varepsilon(\pi_\text{E})$ if and only if, $c \in \text{Lip}(\mathcal{X} \times \mathcal{A})$ and there exists $u \in \text{Lip}(\mathcal{X})$ such that*

$$\begin{cases} \langle \mu_{\nu_0}^{\pi_\text{E}}, c - T_\gamma^* u \rangle & \leq \varepsilon, \\ c - T_\gamma^* u & \geq -\varepsilon, \text{ on } \mathcal{X} \times \mathcal{A}. \end{cases} \tag{1}$$

We are now ready to characterize the solutions to IRL, following arguments from [46, 47, 48].

**Theorem 3.1** (Inverse feasibility set characterization)**.** *Let $\pi_\text{E} \in \Pi$. Under Assumption 2.1 on the Markov decision model $\mathcal{M}_c$, the following assertions are equivalent*

1. *$c \in \mathcal{C}^0(\pi_\text{E})$;*

2. *$c \in \bigcap_{\varepsilon > 0} \mathcal{C}^\varepsilon(\pi_\text{E})$;*

3. $\pi_{\mathrm{E}}$ is $\gamma$-discount $\nu_0$-optimal for ($\mathrm{MDP_c}$) with cost function $c$.

As a consequence, $\mathcal{C}(\pi_{\mathrm{E}}) = \mathcal{C}^0(\pi_{\mathrm{E}}) = \bigcap_{\varepsilon > 0} \mathcal{C}^\varepsilon(\pi_{\mathrm{E}})$. Moreover, $\mathcal{C}(\pi_{\mathrm{E}})$ is a convex cone and $\|\cdot\|_{\mathrm{L}}$-closed in $\mathrm{Lip}(\mathcal{X} \times \mathcal{A})$.

As a result, a cost function is inverse feasible for $\mathcal{B} = (\mathcal{M}, \pi_{\mathrm{E}})$ if and only if it is $\varepsilon$-inverse feasible for all $\varepsilon > 0$. The characterization of the inverse feasibility set is due to linear duality and complementary slackness conditions. In particular, the constraint that holds pointwise in (19) is due to dual feasibility while the constraint that holds in expectation is due to strong duality, The details are provided in the proof of Theorem 3.1 in Appendix B.1.

Notably, when $\mathcal{X}$ and $\mathcal{A}$ are finite, and the expert policy $\pi_{\mathrm{E}}$ is stationary Markov, our formulation aligns with the finite-dimensional inverse feasibility set introduced in [52, 53, 54]. Furthermore, when the expert is deterministic of the form $\pi_{\mathrm{E}}(x) \equiv a_1$, for all $x$, then we recover the linear programs discussed in [12, 49, 51] (see Appendix A.2).

Using occupancy measures instead of policies, we can assess inverse feasibility for continuous MDPs, regardless of expert policy complexity. This approach allows us to utilize offline expert demonstrations for computing an approximate feasibility set and deriving costs via a sample-based convex program This flexibility surpasses previous theoretical settings, where either $\pi_{\mathrm{E}}$ is assumed to be fully known and deterministic [49, 51, 52] or active querying of $\pi_{\mathrm{E}}$ is possible for each state [53, 54].

**Proposition 3.1** ($\varepsilon$-inverse feasibility set characterization)**.** *Under Assumption 2.1, for any $\varepsilon > 0$, it holds that a cost function $\tilde{c}$ is in $\mathcal{C}^\varepsilon(\pi_{\mathrm{E}})$ if and only if $\pi_{\mathrm{E}}$ is $\frac{2-\gamma}{1-\gamma}\varepsilon$-optimal for ($\mathrm{MDP_{\tilde{c}}}$) with cost $\tilde{c}$.*

As $\varepsilon \to 0$, the next proposition indicates a close approximation to the inverse problem solution.

**Proposition 3.2.** *Let $(\varepsilon_n)_n$ be a sequence such that $\lim_{n \to \infty} \varepsilon_n = 0$ and let $c_n \in \mathcal{C}^{\varepsilon_n}(\pi_{\mathrm{E}})$. Then, every accumulation point $c$ of the sequence $(c_n)_n$ is inverse feasible, i.e., $c \in \mathcal{C}(\pi_{\mathrm{E}})$.*

Finally we show that the $\varepsilon$-inverse feasibility set $\mathcal{C}^\varepsilon(\pi_{\mathrm{E}})$ satisfies the $\varepsilon$-optimality criterion considered in [52, 53, 54]; see for example [53, Def. 2].

**Proposition 3.3.** *Let $\varepsilon > 0$. It holds that $\inf_{c \in \mathcal{C}(\pi_{\mathrm{E}})} V_c^{\tilde{\pi}}(\nu_0) - V_c^{\pi_{\mathrm{E}}}(\nu_0) \leq \frac{2-\gamma}{1-\gamma}\varepsilon$, for all $\tilde{c} \in \mathcal{C}^\varepsilon(\pi_{\mathrm{E}})$, where $\tilde{\pi}$ is an optimal policy for the recovered cost $\tilde{c}$.*

This condition ensures that when $\varepsilon$ is small we avoid an unnecessarily large *approximate* feasibility set since there is a possible true cost in $\mathcal{C}(\pi_{\mathrm{E}})$ with a small error for every possible recovered cost function in $\mathcal{C}^\varepsilon(\pi_{\mathrm{E}})$. [3]

## 4  Towards recovering a nearly optimal cost function

Although we characterized the inverse and $\varepsilon$-inverse feasibility sets in Theorem 3.1 and Proposition 3.1 respectively, it is not clear yet how to compute them, as (19) is an infinite-dimensional feasibility LP. In practice, the following challenges need to be addressed:

(a) The inverse problem is ill-conditioned and ill-posed since each task is consistent with many cost functions, and thus a central challenge is to end up with a meaningful one. To avoid trivial solutions, in Section 4.1 we motivate the addition of a linear normalization constraint.

(b) Another challenge appears because the LP formulation is infinite-dimensional, hence computationally intractable. In Section 4.2 we address this problem by proposing a tractable approximation method with probabilistic performance bounds.

(c) In practice, complete knowledge of $\pi_{\mathrm{E}}$ and $\mathrm{P}$ is often unavailable. In Section 4.3, we tackle this challenge by assuming that we have access to a finite set of traces of the expert policy and a *generative-model oracle*. We use empirical counterparts of $\pi_{\mathrm{E}}$ and $\mathrm{P}$ and provide error bounds to quantify our approach's accuracy with sample data.

The main building blocks of our methodology are depicted in Figure 2.

---

[3]The second condition in [53, Def. 2] is trivially satisfied with zero error since $\mathcal{C}(\pi_{\mathrm{E}}) \subset \mathcal{C}^\varepsilon(\pi_{\mathrm{E}})$.

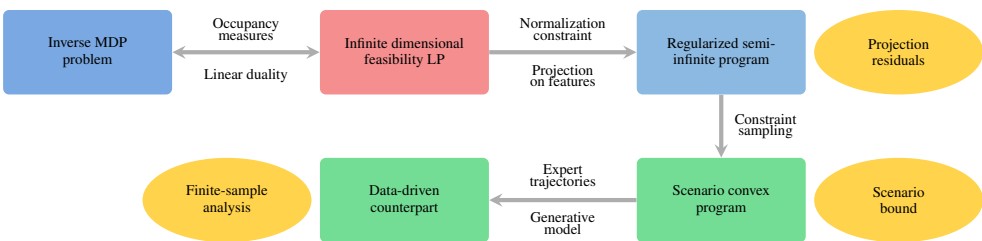

Figure 2: Main building blocks of our methodology

## 4.1 Normalization constraint

A well-known limitation of IRL is that it suffers from the *ambiguity* issue, i.e., the problem admits infinitely many solutions. For example, any constant cost function, including the zero cost, is inverse feasible. In addition, as it is apparent from the characterization of $\mathcal{C}(\pi_E) = \mathcal{C}^0(\pi_E)$ in Theorem 3.1, for any $u \in \text{Lip}(\mathcal{X})$ and $c \in \mathcal{C}(\pi_E)$, the cost $c + T_\gamma^* u$ is inverse feasible. This phenomenon, also known as *reward shaping* [63] refers to the modification or design of a reward function to provide additional guidance or incentives to an agent during learning. In addition, since $\mathcal{C}(\pi_E)$ is a convex cone and closed for the sup-norm (Theorem 3.1) the set of solutions to IRL is closed to convex combinations and uniform limits.

All these examples illustrate that the inverse feasibility set $\mathcal{C}(\pi_E)$ contains some solutions that arise from mathematical artifacts. To mitigate this difficulty and avoid trivial solutions, inspired by [48], we enforce the additional natural normalization constraint $\int_{\mathcal{X} \times \mathcal{A}} (c - T_\gamma^* u)(x, a) \, \mathrm{d}(x, a) = 1$. The following proposition justifies this choice.

**Proposition 4.1.** *If Definition 3.2 of $\mathcal{C}(\pi_E) = \mathcal{C}^0(\pi_E)$ includes the normalization constraint $\int_{\mathcal{X} \times \mathcal{A}} (c - T_\gamma^* u) \mathrm{d}x \mathrm{d}a = 1$, then all constant cost functions are excluded from the inverse feasibility set.*

Alternatively, it is possible to employ additional heuristics to narrow down the set of possible solutions and incorporate prior knowledge, e.g., by restricting the class where the true cost belongs, constraining the dependence between costs and value functions, and enforcing conic constraints or shape conditions.

It is worth mentioning that the normalization constraint in our formulation primarily aims to mitigate the ill-posedness, or ambiguity issue, intrinsic to the IRL problem, rather than to resolve issues of identifiability. In particular, we state and prove that the normalization constraint rules out a wide class of trivial solutions, i.e., all constant functions and inverse solutions of the form $c = T_\gamma^* u$, an outcome devoid of physical meaning and a mathematical artifact. While the identifiability problem and the ill-posedness problem are related in IRL, they are not the same. Identifiability deals with the uniqueness of the true cost function and cannot be fully resolved unless, for example, one has access to multiple expert policies or environments for comparison [64, 65]. On the other hand, ill-posedness is a broader concept from mathematical and statistical problems and refers to situations where a problem does not satisfy the conditions for being well-posed, e.g., in our case due to infinitely many solutions. Note that, unlike recent theoretical IRL works [52, 53, 54] which avoid discussing this issue altogether, we attempt to address the ambiguity problem and provide theoretical results in this direction, hoping to lay the foundations for overcoming current limitations.

## 4.2 The case of known dynamics and expert policy

We first consider the case where the expert policy $\pi_E$, the induced occupation measure $\mu_{\nu_0}^{\pi_E}$ and the transition law $P$ are known. We leverage developments in randomized convex optimization, leading to an approximation scheme with a priori performance guarantees. As a first step, we introduce a semi-infinite convex formulation that enforces the normalization constraint, involves a restriction of the decision variables from an infinite-dimensional function space to a finite-dimensional subspace, and considers an additional norm constraint that effectively acts as a regularizer. The resulting

regularized semi-infinite *inner approximation*, which we call *inverse program* is given by

$$\begin{cases} \inf_{\alpha,\beta,\varepsilon} & \varepsilon \\ \text{s.t.} & \langle \mu_{\nu_0}^{\pi_E}, c - T_\gamma^* u \rangle \leq \varepsilon, \\ & c(x,a) - T_\gamma^* u(x,a) \geq -\varepsilon, \ \forall (x,a) \in \mathcal{X} \times \mathcal{A}, \\ & \int_{\mathcal{X} \times \mathcal{A}} (c - T_\gamma^* u)(x,a) \, d(x,a) = 1, \\ & c \in \mathbf{C}_{n_c}, u \in \mathbf{U}_{n_u}, \varepsilon \geq 0, \end{cases} \tag{IP}$$

where $\mathbf{C}_{n_c} \triangleq \{\sum_{j=1}^{n_c} \alpha_j c_j : \alpha = \{\alpha_i\}_{i=1}^{n_c} \in \mathbb{R}^{n_c}, \|\alpha\|_1 \leq \theta\}$ with $\{c_i\}_{i=1}^{n_c} \subset \text{Lip}(\mathcal{X} \times \mathcal{A})$ being linearly independent basis functions with Lipschitz constant $L_c > 0$, $\mathbf{U}_{n_u} \triangleq \{\sum_{i=1}^{n_u} \beta_i u_i : \beta = \{\beta_i\}_{i=1}^{n_u} \in \mathbb{R}^{n_u}, \|\beta\|_1 \leq \theta\}$, with $\{u_i\}_{i=1}^{n_u} \subset \text{Lip}(\mathcal{X})$ being linearly independent basis functions with Lipschitz constant $L_u > 0$, and $\theta > 0$ is an appropriately chosen regularization parameter. Note that (IP) is derived by relaxing the constraints in the inverse feasibility set $\mathcal{C}(\pi_E)$ and paying a penalty when violated. In particular, let $(\tilde{\varepsilon}, \tilde{\alpha}, \tilde{\beta})$ be an optimal solution of the semi-infinite program (IP). Then, $\tilde{c} \triangleq \sum_{i=1}^{n_c} \tilde{\alpha}_i c_i \in \mathcal{C}^{\tilde{\varepsilon}}(\pi_E)$, from where it becomes apparent that the smaller the value of $\tilde{\varepsilon}$, the better the quality of the extracted cost function $\tilde{c}$ (as by Proposition 3.1). One would intuitively expect that $\tilde{\varepsilon}$ depends on the choice of basis functions for the cost (resp. value) function as well as on the parameters $n_c$ (resp. $n_u$) and $\theta$. This dependency is highlighted by the following proposition.

**Proposition 4.2** (Basis function dependency). *Let $\pi_E$ be an optimal policy for the optimal control problem $\text{MDP}_{c^\star}$ with cost $c^\star$ and let $u^\star$ be the corresponding optimal value function. Under Assumption 2.1, if $u_1 \equiv 1$ and $\theta > \frac{1}{(1-\gamma)\min\{1,d\}}$, then $\tilde{\varepsilon} \leq \varepsilon_{\text{approx}}$ with*

$$\varepsilon_{\text{approx}} := \left( \frac{2-\gamma}{1-\gamma} + \mathcal{D}_{\gamma,\theta}(2+\gamma)\max\{1, L_P, d\} \right) \left( \min_{c \in \mathbf{C}_{n_c}} \|c^\star - c\|_L + \min_{u \in \mathbf{U}_{n_u}} \|u^\star - u\|_L \right), \tag{2}$$

*where $d = \text{leb}(\mathcal{X} \times \mathcal{A})$ is the Lebesgue measure of $\mathcal{X} \times \mathcal{A}$, $\mathcal{D}_{\gamma,\theta} \triangleq \frac{2\theta(K_{c,\infty} + K_{u,\infty})}{(1-\gamma)^2 \min\{1,d\}\theta + \gamma - 1}$, with constants $K_{c,\infty} \triangleq \max_{i=1,\dots,n_c} \|c_i\|_\infty$ and $K_{u,\infty} \triangleq \max_{j=1,\dots,n_u} \|u_i\|_\infty$.*

Proposition 4.2 sheds light on how the choice of basis functions and the regularization parameter $\theta$ influence the approximation error. Essentially, the approximation error term $\varepsilon_{\text{approx}}$ measures the expressiveness of the linear function approximators. Prior knowledge about the properties of the true cost allows us to choose appropriate basis functions to make the projection residuals in the theorem sufficiently small. For example, if the true cost function is known to be smooth, Fourier or polynomial basis functions can be used. In general, if we choose linearly dense bases in $\text{Lip}(\mathcal{X} \times \mathcal{A})$ and $\text{Lip}(\mathcal{X})$, then the projection residuals and so $\tilde{\varepsilon}$ tend to 0 as $n_c$ and $n_u$ and the regularization parameter $\theta$ tend to infinity. In particular note that when $c^\star \in \mathbf{C}_{n_c}$ and $u^\star \in \mathbf{U}_{n_u}$, then the corresponding projection residuals are 0, and thus $\tilde{\varepsilon} = 0$ as expected. In a practical setting, observing a large value of $\tilde{\varepsilon}$ is an indicator that more basis functions are needed.

Finally, note that the regularizer helps to bound the dual optimizer using a dual norm, thus offering an explicit approximation error for the proposed solution (see Appendix B.6).

Computationally tractable approximations to the semi-infinite convex program can be obtained through the *scenario approach* [29, 66] in which randomization over the set of constraints is considered. In particular, we treat the parameter $(x,a)$ as an uncertainty parameter living in the space $\mathcal{X} \times \mathcal{A}$. Let $\mathbb{P}$ be a Borel probability measure on $(\mathcal{X} \times \mathcal{A}, \mathcal{B}(\mathcal{X} \times \mathcal{A}))$, where $\mathcal{X} \times \mathcal{A}$ is equipped with the norm $\|(x,a)\| = \|x\|_2 + \|a\|_2$. We assume that $\mathbb{P}$ has the following structure.

**Assumption 4.1** (Sampling distribution). *There exists $g : \mathbb{R}_+ \to [0,1]$ strictly increasing, such that $\mathbb{P}(B_r(x,a)) > g(r)$, for all $(x,a) \in \mathcal{X} \times \mathcal{A}$ and $r > 0$.*

Assumption 4.1 is a sufficient structural assumption concerning the sample distribution $\mathbb{P}$ ensuring that the gap between the robust program (IP) and its sampled counterpart (SIP$_N$) can be controlled. It implicitly restricts the state and action spaces to be bounded.

Let $\{(x^{(\ell)}, a^{(\ell)})\}_{\ell=1}^N$ be independent and identically distributed (i.i.d.) samples drawn from $\mathcal{X} \times \mathcal{A}$ according to $\mathbb{P}$. We are interested in the following random finite-dimensional convex program:

$$\begin{cases} \inf_{\alpha,\beta,\varepsilon} & \varepsilon \\ \text{s.t.} & \langle \mu_{\nu_0}^{\pi_E}, c - T_\gamma^* u \rangle \leq \varepsilon, \\ & c(x^{(\ell)}, a^{(\ell)}) - T_\gamma^* u(x^{(\ell)}, a^{(\ell)}) \geq -\varepsilon, \ \forall \ell = 1, \dots N, \\ & \int_{\mathcal{X} \times \mathcal{A}} (c(x,a) - T_\gamma^* u(x,a)) \, d(x,a) = 1, \\ & c \in \mathbf{C}_{n_c}, u \in \mathbf{U}_{n_u}, \varepsilon \geq 0. \end{cases} \tag{SIP$_N$}$$

Notice that (SIP$_{\mathbf{N}}$) naturally represents a randomized program as it depends on the random multi-sample $\{(x^{(i)}, a^{(i)})\}_{i=1}^{N}$. We assume the following measurability assumption holds for our analysis.

**Assumption 4.2.** *The* (SIP$_{\mathbf{N}}$) *optimizer generates a Borel measurable mapping that associates each multi-sample* $\{(x^{(\ell)}, a^{(\ell)})\}_{\ell=1}^{N}$ *to a uniquely defined optimizer* $(\tilde{\alpha}_N, \tilde{\beta}_N, \tilde{\varepsilon}_N)$.

To ensure uniqueness when multiple solutions exist, use a *tie-break rule*, such as selecting the solution with the minimum $\|\cdot\|_2$ norm. It has been shown [30, Proposition 3.10] that applying such a tie-break-rule also ensures the measurability in Assumption 4.2.

The appealing feature of (SIP$_{\mathbf{N}}$) is that it is a convex finite-dimensional program and so it can be solved at low computational cost for small enough $N$. We study how many samples are needed for a *good solution* by examining the *generalization properties* of the optimizer $(\tilde{\alpha}_N, \tilde{\beta}_N, \tilde{\varepsilon}_N)$ and the extracted cost function $\tilde{c}N = \sum_{i=1}^{n_c} \tilde{\alpha}N_i c_i$. For each $n \in \mathbb{N}$, $\epsilon \in (0, 1)$ and $\delta \in (0, 1)$, we define

$$\mathrm{N}(n, \epsilon, \delta) = \min\left\{ N \in \mathbb{N} : \sum_{i=0}^{n} \binom{N}{i} \epsilon^i (1-\epsilon)^{N-i} \leq \delta \right\}.$$

The sample size above asymptotically scales as $\sim \{1/\epsilon, \; \log(1/\delta), \; n\}$, see [29]. The following theorem determines the sample complexity of (SIP$_{\mathbf{N}}$). In particular, for a given reliability parameter $\epsilon \in (0, 1)$ and confidence level $\delta \in (0, 1)$, we establish how many samples are needed to guarantee with confidence at least $1 - \delta$ that $\tilde{c}_N \in \mathcal{C}^{\varepsilon_{\text{approx}} + \epsilon}(\pi_{\mathrm{E}})$.

**Theorem 4.1** (Scenario program guarantees)**.** *Let* $\epsilon, \delta \in (0, 1)$. *Under Assumptions 2.1, 4.1 and 4.2, if* $u_1 \equiv 1$ *and* $\theta > \frac{1}{(1-\gamma)\mathrm{leb}(\mathcal{X} \times \mathcal{A})}$, *then by sampling* $N \geq \mathrm{N}(n_c + n_u + 1, g(\frac{\epsilon}{L_\Lambda}), \delta)$ *constraints, where* $L_\Lambda \triangleq \theta\sqrt{n_c}L_c + \theta\sqrt{n_u}(L_u L_{\mathsf{P}} + L_u)$, *with probability at least* $1 - \delta$, $\tilde{c}_N \in \mathcal{C}^{\varepsilon_{\text{approx}} + \epsilon}(\pi_{\mathrm{E}})$.

**Remark 4.1** (Curse of dimensionality)**.** As shown in [30], the function $g(r)$ is of order $r^{\dim(\mathcal{X} \times \mathcal{A})}$. As a result, the number of samples grows exponentially as $\epsilon^{-\dim(\mathcal{X} \times \mathcal{A})}$. A similar exponential dependence to the dimension of the state space has been established in [51]. Considering the current performance of general LP solvers, this approach is attractive for small to medium-sized problems. As noted in [51], dealing with the general $d$-dimensional case without exponential scaling in $d$ is challenging. Therefore, understanding the selection of a suitable distribution for future sample drawing is crucial.

**Remark 4.2** ($l_1$-regularization)**.** To cut down on required samples $N$, a common method is using $l_1$-regularization to reduce the effective dimension of the optimization variable. This concept is formalized in the current setting [67]. Moreover, in the spirit of the compressed sensing literature [68], $l_1$-regularization will promote sparse solutions and hence lead to "simple" cost functions. In the context of optimal control $l_1$-regularization term is studied as the so-called "maximum hands-off control" paradigm [69, 70]. In our case, the utilization of the $l_1$-norm offers two primary advantages. Firstly, the $l_1$-norm promotes sparsity in solutions, thereby potentially reducing computational demands. Secondly, this specific type of regularization preserves the linearity of the program.

## 4.3   Sample-based inverse reinforcement learning

In this section, we explore the realistic scenario where we lack access to the expert policy $\pi_{\mathrm{E}}$ and the transition law $\mathsf{P}$. The learner only receives a finite set of truncated expert sample trajectories and cannot interact or query the expert for additional data during training. Despite the unknown MDP model, we assume access to a *generative-model oracle*. It provides the next state $x'$ given a state-action pair $(x, a)$ sampled from $\mathsf{P}(\cdot | x, a)$. This is known as the simulator-defined MDP [71, 72].

**Sampling process.**   Let $\tau = \{\tau_i = (x_0^i, a_0^i, \ldots, x_H^i, a_H^i)\}_{i=1}^{m}$ be i.i.d., truncated sample trajectories according to $\pi_{\mathrm{E}}$. For any $c \in \mathrm{Lip}(\mathcal{X} \times \mathcal{A})$, we consider the sample average approximation of the expectation $\langle \mu_{\nu_0}^{\pi_{\mathrm{E}}}, c \rangle$ given by, $\widehat{\langle \mu_{\nu_0}^{\pi_{\mathrm{E}}}, c \rangle}(\tau) \triangleq \frac{1}{m} \sum_{t=0}^{H} \sum_{j=1}^{m} \gamma^t c(x_t^j, a_t^j)$. Similarly, if $\xi = \{x_0^k\}_{k=1}^{n}$ are i.i.d., samples according to $\nu_0$, for any $u \in \mathrm{Lip}F(\mathcal{X})$, we define the corresponding sample average estimation of the expectation $\langle \nu_0, u \rangle$ by $\widehat{\langle \nu_0, u \rangle}(\xi) \triangleq \frac{1}{n} \sum_{k=1}^{n} u(x_0^k)$. Moreover, let $\zeta = \{(x^{(l)}, a^{(l)})\}_{l=1}^{N}$ be i.i.d. samples drawn from $\mathcal{X} \times \mathcal{A}$ according to $\mathbb{P} \in \mathcal{P}(\mathcal{X} \times \mathcal{A})$. We also use the following notation $\widehat{T}_\gamma^* u(x^{(l)}, a^{(l)}, y^{(l)}) \triangleq u(x^{(l)}) - \frac{1}{k} \sum_{i=1}^{k} u(y_i^{(l)})$, $\{y_i^{(l)}\}_{i=1}^{k} \overset{\text{i.i.d.}}{\sim} \mathsf{P}(\cdot | x^{(l)}, a^{(l)})$.

We are interested in the finite-sample analysis of the following random convex program:

$$
\begin{cases}
\inf_{\alpha,\beta,\varepsilon} & \varepsilon \\
\text{s.t.} & \widehat{\langle\mu_{\nu_0}^{\pi_E}, c\rangle}(\tau) - \widehat{\langle\nu_0, u\rangle}(\xi) \le \varepsilon, \\
& c(x^{(l)}, a^{(l)}) - \widehat{T}_\gamma^* u(x^{(l)}, a^{(l)}, y^{(l)}) \ge -\varepsilon, \ \forall l = 1,\dots N, \\
& \alpha \in \Delta_{[n_u]}, \beta \in \Delta_{[n_c]}, \\
& c \in \mathbf{C}_{n_c}, u \in \mathbf{U}_{n_u}, \varepsilon \ge 0,
\end{cases}
\qquad \text{(SIP}_{\mathbf{N,m,n,k}})
$$

where the function classes $\mathbf{C}_{n_c}, \mathbf{U}_{n_u}$ are defined as in the previous Section. We make the following measurability assumption which is analogous to Assumption 4.2.

**Assumption 4.3.** *The* (SIP$_{\mathbf{N,m,n,k}}$) *optimizer generates a Borel measurable mapping that associates each multi-sample* $(y, \tau, \xi, \zeta)$ *to a uniquely defined optimizer* $(\tilde{\alpha}_{N,m,n,k}, \tilde{\beta}_{N,m,n,k}, \tilde{\varepsilon}_{N,m,n,k})$.

**Theorem 4.2.** *Under Assumptions 2.1, 4.1 and 4.3, if* $u_1 \equiv 1$ *and* $\theta > \frac{1}{(1-\gamma)\text{leb}(\mathcal{X}\times\mathcal{A})}$, *then for* $N \ge \text{N}(n_c + n_u + 1, g(\frac{\epsilon}{L_\Lambda}), \delta/2)$ *constraints,* $n \ge \frac{8K_{u,\infty}^2 \theta^2 n_u \ln(\frac{8n_u}{\delta})}{\epsilon^2}$, $m \ge \frac{8K_{c,\infty}^2 \theta^2 n_c \ln(\frac{8n_c}{\delta})}{(1-\gamma)^2\epsilon^2}$ *expert samples with horizon* $H = \frac{1}{1-\gamma}\log(\frac{2}{\varepsilon})$, *and* $k \ge \frac{8Cn_u\theta^2 \log(4n_uN/\gamma)}{\epsilon^2}$ *calls to the generative model per constraint, with probability at least* $1-\delta$, $\tilde{c}_{N,m,n,k} \in \mathcal{C}^{\varepsilon_{\text{approx}}+\epsilon}(\pi_E)$. *The constants* $K_{c,\infty}$ *and* $K_{u,\infty}$ *are given as* $K_{c,\infty} \triangleq \max_{i=1,\dots,n_c} \|c_i\|_\infty$ *and* $K_{u,\infty} \triangleq \max_{j=1,\dots,n_u} \|u_i\|_\infty$.

Theorem 4.2 provides explicit sample complexity bounds for achieving a desired approximation accuracy with our proposed randomized algorithm. For continous MDPs when we use $n_u$ basis functions for the value function and $n_c$ basis functions for the cost function we need $m = \mathcal{O}\left(\frac{n_c \log(\frac{n_c}{\delta})}{(1-\gamma)^2\varepsilon^2}\right)$ expert samples and $K = \mathcal{O}\left(\frac{n_u \log(\frac{n_uN}{\delta})}{\varepsilon^2}\right)$ calls to the generative model per constraint and $N = \mathcal{O}(\exp^{dim(X\times A)})$ sampled constraints and solve the resulted sampled finite LP with $n_u + n_c$ variables and $N$ constraints in polynomial time to learn a cost that is $\varepsilon + \varepsilon_{\text{approx}}$-inverse feasible with probability $1-\delta$.

The corresponding sample complexities include the expert sample complexity $m$, the number of calls to the generator per constraint $k$, and the number of sampled constraints $N$. The first two complexities scale gracefully with respect to the problem parameters, whereas the number of sampled constraints scales exponentially with the dimension of the state and action spaces (see also Remark 4.1). This makes the algorithm particularly suitable for low-dimensional problems of practical interest, e.g., pendulum swing-up control, vehicle cruise control, and quadcopter stabilization. Note that a similar exponential dependence to the dimension of the state space has been established in Dexter et al. [51], a theoretical study addressing IRL in continuous state but discrete action spaces.

A promising research direction is to enhance the sample complexity bounds through the utilization of the underlying problem structure [73]. In addition, it becomes imperative to gain an understanding regarding the selection of a suitable distribution for drawing samples in the future. Intuitively, it is reasonable to anticipate that certain regions within the state-action space carry more "informative" characteristics than others. One conjecture is that sampling constraints based on the expert occupancy measure could yield a more scalable bound [74]. However, a comprehensive mathematical treatment of these inquiries will be addressed in future research endeavors.

**Redunction to tabular MDPs.** For tabular MDPs, offline access to expert for tabular MDPs, and a generative model we require $m = \mathcal{O}\left(\frac{|X||A|(\log(\frac{|X||A|}{\delta})}{(1-\gamma)^2\varepsilon^2}\right)$ expert samples, and $K|X||A| = \mathcal{O}\left(\frac{X|^2|A|\log(\frac{|X|^2|A|}{\delta})}{\varepsilon^2}\right)$ calls to the generative model, and solve the resulted sampled finite LP with $|XA| + |X|$ variables and $|X||A| + 2$ constraints in polynomial time, to learn a cost function that is $\varepsilon$-inverse feasible with probability $1-\delta$.

Note that as we argued in detail above our setting is different from the one in [52-54] since we have offline access to the expert and address a different question, i.e. learning a single reward with formal guarantees in continuous MDPs. In this case, there is no need to solve the resulting linear program.

**Numerical Experiment.** In Appendix C, we illustrate our method with a simple truncated Linear Quadratic Regulator (LQR) example to provide better intuition about the method and the proposed sample complexity bounds.

## Acknowledgments

This work was supported by the DFG in the Cluster of Excellence EXC 2117 "Centre for the Advanced Study of Collective Behaviour" (Project-ID 390829875).

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

# A  Supplementary material

## A.1  The linear programming approach for continuous MDPs

In this section, we present essential facts and derivations concerning the Linear Programming (LP) approach to continuous Markov Decision Processes (MDPs). These insights and results will serve as valuable foundations for the subsequent discussions and analysis.

The following theorem summarizes the properties of the optimal value function under the assumption that the control model is Lipschitz continuous.

**Theorem A.1** ([60, Theorem 3.1], [61]). *Under Assumption 2.1, the following hold:*

1. *The value function $V_c^\star$ is in $\mathrm{Lip}(\mathcal{X})$ with Lipschitz constant $L_{V_c^\star} \leq L_c + \frac{\gamma}{1-\gamma}\|c\|_\infty L_{\mathsf{P}}$;*

2. *The value function $V_c^\star$ satisfies the* Bellman optimality equation
$$V_c^\star(x) = \min_{a\in\mathcal{A}}\{c(x,a) + \gamma \int_{\mathcal{X}} V_c^\star(y)Q(dy|x,a)\}, \quad \text{for all } x \in \mathcal{X};$$

3. *There exists a $\gamma$-discount $\nu_0$-optimal policy which is stationary deterministic.*

Next, we characterize the set of occupation measures in terms of linear constraint satisfaction.

**Theorem A.2** ([62, Theorem 6.3.7]). *Consider the convex set of measures, $\mathfrak{F} \triangleq \{\mu \in \mathcal{M}(\mathcal{X} \times \mathcal{A})_+ : T_\gamma \mu = \nu_0\}$, where $T_\gamma : \mathcal{M}(\mathcal{X} \times \mathcal{A}) \to \mathcal{M}(\mathcal{X})$ is a linear and weakly continuous operator given by*

$$(T_\gamma\mu)(B) \triangleq \mu(B \times \mathcal{A}) - \gamma \int_{\mathcal{X}\times\mathcal{A}} \mathsf{P}(B|x,a)\mu(\mathrm{d}(x,a)), \quad \text{for all } B \in \mathcal{B}(\mathcal{X}).$$

*Then, $\mathfrak{F} = \{\mu_{\nu_0}^\pi : \pi \in \Pi_0\}$. Moreover, $\langle \mu_{\nu_0}^\pi, c \rangle = V_c^\pi(\nu_0)$, for every $\pi$.*

A direct consequence of Theorem A.2 is that

$$V_c^\star(\nu_0) = \min_\pi \langle \mu_{\nu_0}^\pi, c \rangle = \min_{\pi\in\Pi_0} \langle \mu_{\nu_0}^\pi, c \rangle = \inf_{\mu\in\mathfrak{F}} \langle \mu, c \rangle .$$

Therefore the MDP problem ($\mathrm{MDP_c}$) can be stated equivalently as an infinite-dimensional LP over measures

$$\mathcal{J}_c(\nu_0) \triangleq \inf_{\mu\in\mathcal{M}(\mathcal{X}\times\mathcal{A})_+} \{\langle\mu,c\rangle : T_\gamma\mu = \nu_0\}. \tag{$\mathrm{P_c}$}$$

In particular the infimum in ($\mathrm{P_c}$) is attained and $\pi^\star$ is optimal for the OCP ($\mathrm{MDP_c}$) if and only if $\mu_{\nu_0}^{\pi^\star}$ is optimal for the primal LP ($\mathrm{P_c}$).

Consider the dual pair of vector spaces $(\mathcal{M}(\mathcal{X}\times\mathcal{A}), \mathrm{Lip}(\mathcal{X}\times\mathcal{A}))$ and $(\mathcal{M}(\mathcal{X}), \mathrm{Lip}(\mathcal{X}))$. Then the adjoint linear operator $T_\gamma^* : \mathrm{Lip}(\mathcal{X}) \to \mathrm{Lip}(\mathcal{X}\times\mathcal{A})$ of $T_\gamma$ is given by

$$(T_\gamma^*u)(x,a) \triangleq u(x) - \gamma \int_{\mathcal{X}} u(y)\mathsf{P}(\mathrm{d}y|x,a).$$

Indeed, $T_\gamma^*$ is well-defined by Assumption (A2). Moreover, a direct computation shows that (see [28, Pg. 139])

$$\langle \mu, T_\gamma^*u \rangle = \langle T_\gamma\mu, u \rangle , \quad \text{for all } \mu \in \mathcal{M}(\mathcal{X}\times\mathcal{A}), \ u \in \mathrm{Lip}(\mathcal{X}).$$

In addition, the dual convex cone of $\mathcal{M}(\mathcal{X}\times\mathcal{A})_+$ is the set $\mathrm{Lip}(\mathcal{X}\times\mathcal{A})_+$ of nonnegative bounded and Lipschitz continuous functions on $\mathcal{X}\times\mathcal{A}$. This is because,

$$\begin{aligned}\mathcal{M}(\mathcal{X}\times\mathcal{A})_+ &\triangleq \{v \in \mathrm{Lip}(\mathcal{X}\times\mathcal{A}) \mid \langle\mu,v\rangle \geq 0, \ \forall \ \mu \in \mathcal{M}(\mathcal{X}\times\mathcal{A})_+\} \\ &= \{v \in \mathrm{Lip}(\mathcal{X}\times\mathcal{A}) \mid v \geq 0\}.\end{aligned}$$

The dual LP of ($\mathrm{P_c}$) is given by

$$\mathcal{J}_c^*(\nu_0) \triangleq \sup_{u\in\mathrm{Lip}(\mathcal{X})} \{\langle\nu_0,u\rangle : c - T_\gamma^*u \geq 0 \text{ on } \mathcal{X}\times\mathcal{A}\}. \tag{$\mathrm{D_c}$}$$

**Theorem A.3** (Strong duality). *Under Assumption 2.1 on the control model $\mathcal{M}_c$, the dual LP ($D_c$) is solvable, i.e., the supremum is attained, and strong duality holds. That is, $\mathcal{J}_c(\nu_0) = \mathcal{J}_c^*(\nu_0) = V_c^\star(\nu_0)$. In particular the value function $V_c^\star$ is an optimal solution for the dual LP ($D_c$).*

*Proof.* By virtue of [62, Theorem 6.3.8] we have that
$$V_c^\star(\nu_0) = \mathcal{J}_c(\nu_0) = \mathcal{J}_c^\star(\nu_0)$$
and moreover the supremum definiting $\mathcal{J}_c^\star(\nu_0)$ is attained by the optimal value function $V_c^\star : \mathcal{X} \to \mathbb{R}$. Therefore, the result follows by 1) of Theorem A.1. $\qquad\square$

### A.2 Comparison to the LP formulations for IRL from the literature

In this section, we will demonstrate that our formulation, when applied to finite tabular MDPs and a stationary Markov expert policy $\pi_E$, simplifies to the inverse feasibility set considered in recent studies [52, 53, 54]. The formulation presented in these works extends the LP formulation previously explored in [12, 49, 50, 51], which specifically addressed deterministic expert policies of the form $\pi_E(s) \equiv a_1$. By highlighting this connection, we establish a link between our approach and the existing body of literature on LP formulations for IRL, while also accounting for continuous state and action spaces and more general expert policies.

Let $\mathcal{X}$ and $\mathcal{A}$ be finite sets with cardinality $|\mathcal{X}|$ and $|\mathcal{A}|$, respectively. Then, Assumption 2.1 is trivially satisfied and $\mathrm{Lip}(\mathcal{X} \times \mathcal{A}) = \mathbb{R}^{|\mathcal{X}||\mathcal{A}|}$.

By Theorem 3.1 we have that a cost function $c \in \mathbb{R}^{|\mathcal{X}||\mathcal{A}|}$ is inverse feasible with respect to $\pi_E$, i.e., $c \in \mathcal{C}(\pi_E)$, if and only if there exists $u \in \mathbb{R}^{|\mathcal{X}|}$ such that

$$\begin{cases} \sum_{x,a} \mu_{\nu_0}^{\pi_E}(x,a)(c - T_\gamma^* u)(x,a) = 0, \\ (c - T_\gamma^* u)(x,a) \geq 0, \ \text{ for all } (x,a) \in \mathcal{X} \times \mathcal{A}. \end{cases} \tag{3}$$

If $\pi_E \in \Pi_0$ is a stationary Markov policy, then $\mu_{\nu_0}^{\pi_E}(x,a) = \sum_{a'} \mu_{\nu_0}^{\pi_E}(x,a')\pi_E(a|x)$ and $\sum_{a'} \mu_{\nu_0}^{\pi_E}(x,a') > 0$ [58, Thm. 6.9.1]. Therefore, for any state-action pair $(x,a) \in \mathcal{X} \times \mathcal{A}$, it holds that $\mu_{\nu_0}^{\pi_E}(x,a) = 0 \Leftrightarrow \pi(a|x) = 0$. We then get that (3) is equivalent to

$$\begin{cases} (c - T_\gamma^* u)(x,a) = 0, & \text{if } \pi_E(a|x) > 0 \\ (c - T_\gamma^* u)(x,a) \geq 0, & \text{if } \pi_E(a|x) = 0. \end{cases} \tag{4}$$

So we end up that a cost function $c \in \mathbb{R}^{|\mathcal{X}||\mathcal{A}|}$ is inverse feasible if and only if there exists $u \in \mathbb{R}^{|\mathcal{X}|}$ and $A_\gamma \in \mathbb{R}_+^{|\mathcal{X}||\mathcal{A}|}$ such that for all $(x,a) \in \mathcal{X} \times \mathcal{A}$,

$$c(x,a) - u(x) + \gamma \sum_y \mathsf{P}(y|x,a)u(y) = A_\gamma(x,a)\mathbb{1}_{\{\pi_E(a|x)=0\}}.$$

So we have recovered [52, Lem. 3.2], which forms the basis for the analysis and algorithms in [52, 53, 54].

Next, note that when $\pi_E(x) = a_1$, for all $x \in \mathcal{X}$ and $c(x,a) = c(x)$, for all $(x,a) \in \mathcal{X} \times \mathcal{A}$, then (4) is equivalent to

$$\begin{cases} c(x) - u(x) = -\gamma \sum_y \mathsf{P}(y|x,a_1)u(y), & \text{for all } x \in \mathcal{X} \\ c(x) - u(x) \geq -\gamma \sum_y \mathsf{P}(y|x,a)u(y), & \text{for all } x \in \mathcal{X}, \ a \in \mathcal{A}\backslash\{a_1\}. \end{cases} \tag{5}$$

Therefore, a cost function $c \in \mathbb{R}^{|\mathcal{X}||\mathcal{A}|}$ is inverse feasible if and only if there exists $u \in \mathbb{R}^{|\mathcal{X}|}$ such that

$$\begin{cases} c(x) - u(x) = -\gamma \sum_y \mathsf{P}(y|x,a_1)u(y), & \text{for all } x \in \mathcal{X} \\ \sum_y \mathsf{P}(y|x,a_1)u(y) \leq \sum_y \mathsf{P}(y|x,a)u(y), & \text{for all } x \in \mathcal{X}, \ a \in \mathcal{A}\backslash\{a_1\}. \end{cases} \tag{6}$$

By introducing the notation $\mathsf{P}_a \in \mathbb{R}^{|\mathcal{X}||\mathcal{X}|}$ with $\mathsf{P}_a(x,y) = \mathsf{P}(y|x,a_1)$ and noting that the first linear system in (6) admits a unique solution $u = (\mathsf{I}_{|\mathcal{X}|} - \gamma \mathsf{P}_{a_1})^{-1}c$, we end up that a cost function $c \in \mathbb{R}^{|\mathcal{X}||\mathcal{A}|}$ is inverse feasible if and only if

$$(\mathsf{P}_{a_1} - \mathsf{P}_a)(\mathsf{I}_{|\mathcal{X}|} - \gamma \mathsf{P}_{a_1})^{-1}c \leq 0, \quad \text{for all } a \in \mathcal{A}\backslash\{a_1\}.$$

So we have recovered [12, Thm. 3] which forms the basis for the analysis and algorithms in [12, 49, 50].

# B Proofs

## B.1 Proof of Theorem 3.1

*Proof.* The direction $3) \Rightarrow 1)$ is a consequence of the strong duality Theorem A.3 and complementary slackness. Indeed, assume that $\pi_E$ is optimal for (MDP$_c$) with cost $c$. Then, by Theorem A.2 $\mu_{\nu_0}^{\pi_E}$ is optimal to (P$_c$). By Theorem A.3, the dual (D$_c$) is solvable and there is no duality gap. Therefore, there exists a $u \in \mathrm{Lip}(\mathcal{X})$ such that

$$c - T_\gamma^* u \geq 0, \quad \text{on } \mathcal{X} \times \mathcal{A} \quad \text{(feasibility)},$$
$$\left\langle \mu_{\nu_0}^{\pi_E}, c \right\rangle = \left\langle \nu_0, u \right\rangle \quad \text{(strong duality)}.$$

The second equality is equivalent to $\left\langle \mu_{\nu_0}^{\pi_E}, c - T_\gamma^* u \right\rangle = 0$, since $T_\gamma \mu_{\nu_0}^{\pi_E} = \nu_0$. This proves the desired implication.

The implication $1) \Rightarrow 2)$ is straightforward.

To show $2) \Rightarrow 3)$, let $c \in \bigcap_{\varepsilon > 0} \mathcal{C}^\varepsilon(\pi_E)$. Then, for each $n \in \mathbb{N}$, there exists $u_n \in \mathrm{Lip}(\mathcal{X})$ such that $\left\langle \mu_{\nu_0}^{\pi_E}, c - T_\gamma^* u_n \right\rangle \leq \frac{1}{n}$ and $c - T_\gamma^* u_n \geq -\frac{1}{n}$, on $\mathcal{X} \times \mathcal{A}$. Set $v_n = u_n - \frac{1}{n(1-\gamma)}$. Then, $\{v_n\}_{n=1}^\infty \subset \mathrm{Lip}(\mathcal{X})$ and

$$\lim_{n \to \infty} \left\langle \mu_{\nu_0}^{\pi_E}, c - T_\gamma^* v_n \right\rangle = 0, \tag{7}$$
$$c - T_\gamma^* v_n \geq 0, \quad \text{on } \mathcal{X} \times \mathcal{A}, \tag{8}$$

where we used that $\left\langle \mu_{\nu_0}^{\pi_E}, 1 \right\rangle = \frac{1}{1-\gamma}$ and $T_\gamma^* 1 = 1 - \gamma$, on $\mathcal{X} \times \mathcal{A}$. Equation (8) states that $\{v_n\}_{n=1}^\infty$ is feasible for the dual (D$_c$). Moreover, $\mu_{\nu_0}^{\pi_E}$ is feasible for (P$_c$). Therefore,

$$\left\langle \nu_0, v_n \right\rangle \leq \mathcal{J}_c^*(\nu_0) = \mathcal{J}_c(\nu_0) \leq \left\langle \mu_{\nu_0}^{\pi_E}, c \right\rangle. \tag{9}$$

By (7) $\lim_{n \to \infty} \left\langle \nu_0, v_n \right\rangle = \lim_{n \to \infty} \left\langle T_\gamma \mu_{\nu_0}^{\pi_E}, v_n \right\rangle = \lim_{n \to \infty} \left\langle \mu_{\nu_0}^{\pi_E}, T_\gamma^* v_n \right\rangle = \left\langle \mu_{\nu_0}^{\pi_E}, c \right\rangle$. So, by taking the limits in (9) as $n \to \infty$, we conclude that $\mu_{\nu_0}^{\pi_E}$ is optimal for (P$_c$). Then, by Theorem A.2 $\pi_E$ is optimal to (MDP$_c$) with cost $c$.

Hence, we have shown that $\mathcal{C}(\pi_E) = \bigcap_{\varepsilon > 0} \mathcal{C}^\varepsilon(\pi_E) = \mathcal{C}^0(\pi_E)$. One can check easily that $\mathcal{C}(\pi_E)$ is a convex cone. To show that $\mathcal{C}(\pi_E)$ is $\|\cdot\|_L$-closed in $\mathrm{Lip}(\mathcal{X} \times \mathcal{A})$, let $\{c_n\}_{n=1}^\infty \subset \mathcal{C}(\pi_E)$, such that $\lim_{n \to \infty} \|c_n - c\|_L = 0$, for some $c \in \mathrm{Lip}(\mathcal{X} \times \mathcal{A})$. Let $\varepsilon > 0$. Then, there exists $n_0 \in \mathbb{N}$ such that,

$$\|c_{n_0} - c\|_\infty < \frac{(1-\gamma)\varepsilon}{2}. \tag{10}$$

On the other hand, $c_{n_0} \in \mathcal{C}^{\frac{\varepsilon}{2}}(\pi_E)$. Combining this with (10), we deduce that $c \in \mathcal{C}^\varepsilon(\pi_E)$. Since this is true for arbitrary $\varepsilon > 0$, we get $c \in \bigcap_{\varepsilon > 0} \mathcal{C}^\varepsilon(\pi_E) = \mathcal{C}(\pi_E)$, which proves the desired closedness. □

## B.2 Proof of Proposition 3.1

*Proof.* Assume first that $\tilde{c} \in \mathcal{C}^\varepsilon(\pi_E)$ for some $\varepsilon > 0$. Then, there exists $\tilde{u} \in \mathrm{Lip}(\mathcal{X})$ such that

$$\left\langle \mu_{\nu_0}^{\pi_E}, \tilde{c} - T_\gamma^* \tilde{u} \right\rangle \leq \varepsilon, \tag{11}$$
$$\tilde{c} - T_\gamma^* \tilde{u} \geq -\varepsilon, \quad \text{on } \mathcal{X} \times \mathcal{A}. \tag{12}$$

Since $T_\gamma \mu_{\nu_0}^{\pi_E} = \nu_0$, (11) can be written equivalently as

$$\underbrace{\left\langle \mu_{\nu_0}^{\pi_E}, \tilde{c} \right\rangle}_{=V_{\tilde{c}}^{\pi_E}(\nu_0)} - \left\langle \nu_0, \tilde{u} \right\rangle \leq \varepsilon. \tag{13}$$

Let $\tilde{\mu}$ be an optimal solution to the primal (P$_{\tilde{c}}$) with cost function $\tilde{c}$. By integrating (12) with respect to $\tilde{\mu}$ and using that $T_\gamma \tilde{\mu} = \nu_0$ and $\langle \tilde{\mu}, 1 \rangle = \frac{1}{1-\gamma}$, we get

$$\underbrace{\langle \tilde{\mu}, \tilde{c} \rangle}_{=V_{\tilde{c}}^\star(\nu_0)} - \langle \nu_0, \tilde{u} \rangle \geq \frac{-\varepsilon}{1-\gamma}. \tag{14}$$

Therefore, by combining (13) and (14), we get

$$V_{\tilde{c}}^{\star}(\nu_0) \leq V_{\tilde{c}}^{\pi_{\mathrm{E}}}(\nu_0) \leq V_{\tilde{c}}^{\star}(\nu_0) + \frac{2-\gamma}{1-\gamma}\varepsilon.$$

This proves that $\pi_{\mathrm{E}}$ is $\frac{2-\gamma}{1-\gamma}$-optimal for (MDP$_{\tilde{c}}$) with cost $\tilde{c}$.

For the inverse inclusion, $\pi_{\mathrm{E}}$ be $\frac{2-\gamma}{1-\gamma}$-optimal for (MDP$_{\tilde{c}}$), and let $\hat{u} = V_{\tilde{c}}^{\star} \in \mathrm{Lip}(\mathcal{X})$ (Theorem A.1) be the optimal value function for the forward (MDP$_{\tilde{c}}$). Then, by virtue of Theorem A.3, the following hold:

$$\tilde{c} - T_{\gamma}^{*}\hat{u} \geq 0, \text{ on } \mathcal{X} \times \mathcal{A}, \tag{15}$$

$$\underbrace{\langle \mu_{\nu_0}^{\pi_{\mathrm{E}}}, \tilde{c} \rangle}_{=V_{\tilde{c}}^{\pi_{\mathrm{E}}}(\nu_0)} - \underbrace{\langle \nu_0, \hat{u} \rangle}_{=V_{\tilde{c}}^{\star}(\nu_0)} \leq \frac{2-\gamma}{1-\gamma}\varepsilon. \tag{16}$$

Note that (15) holds due to dual feasibility, while (16) holds because $\pi_{\mathrm{E}}$ is $\frac{2-\gamma}{1-\gamma}$-optimal for (MDP$_{\tilde{c}}$). By setting $\tilde{u} = \hat{u} + \frac{\varepsilon}{1-\gamma} \in \mathrm{Lip}(\mathcal{X})$, we get by (15) that $\tilde{c} - T_{\gamma}^{*}\tilde{u} \geq -\varepsilon$, on $\mathcal{X} \times \mathcal{A}$. Moreover, $\langle \mu_{\nu_0}^{\pi_{\mathrm{E}}}, \tilde{c} - T_{\gamma}^{*}\tilde{u} \rangle = \langle \mu_{\nu_0}^{\pi_{\mathrm{E}}}, \tilde{c} \rangle - \langle \nu_0, \hat{u} \rangle - \langle \nu_0, \frac{\varepsilon}{1-\gamma} \rangle \leq \frac{2-\gamma}{1-\gamma}\varepsilon - \frac{\varepsilon}{1-\gamma} = \varepsilon$, where the inequality holds due to (16). Therefore, $\tilde{c} \in \mathcal{C}^{\varepsilon}(\pi_{\mathrm{E}})$. This concludes the proof. $\qquad \square$

## B.3 Proof of Proposition 3.2

Before stating the proof of Proposition 3.2 we need the following preparation. Without loss of generality, assume that the true cost belongs to $\mathcal{C}_{\mathrm{convex}} \triangleq \{\sum_{i=1}^{k} \alpha_i c_i \mid \alpha \geq 0, \sum_{i=1}^{n} \alpha_i = 1\}$, where $\{c_i\}_{i=1}^{k} \subset \mathrm{Lip}(\mathcal{X} \times \mathcal{A})$ are known features. By linearity the true optimal value function belongs to $\{\sum_{i=1}^{k} \alpha_i u_i \mid \alpha \geq 0, \sum_{i=1}^{k} \alpha_i = 1\}$, where $u_i = V_{c_i}^{\star}(\pi_{\mathrm{E}})$, for all $i = 1, \ldots, k$. Note that by Theorem A.1, $\{u_i\}_{i=1}^{k} \subset \mathrm{Lip}(\mathcal{X})$. By using similar arguments to the proof of Theorem 3.1, we get that a cost function $c$ is inverse feasible, i.e., $c \in \mathcal{C}(\pi_{\mathrm{E}})$ if and only if there exists $\alpha \in \mathbb{R}^k$ such that

$$\begin{cases} \sum_{i=1}^{k} \alpha_i \langle \mu_{\nu_0}^{\pi_{\mathrm{E}}}, c_i - T_{\gamma}^{*}u_i \rangle &= 0, \\ \sum_{i=1}^{k} \alpha_i(c_i - T_{\gamma}^{*}u_i) &\geq 0, \text{ on } \mathcal{X} \times \mathcal{A} \\ \alpha \geq 0, \ \sum_{i=1}^{k} \alpha_i = 1. \end{cases} \tag{17}$$

Similar arguments hold for any choice of finite-dimensional space or convex set $S \subset \mathrm{Lip}(\mathcal{X})$.

*Proof.* Let $\{c_n\}_{n=1}^{\infty} \subset \mathcal{C}^{\varepsilon_n}(\pi_{\mathrm{E}})$. For each $n \in \mathbb{N}$, there exists $\alpha_n \in \mathbb{R}^k$, such that

$$\begin{cases} \sum_{i=1}^{k} \alpha_{n,i} \langle \mu_{\nu_0}^{\pi_{\mathrm{E}}}, c_i - T_{\gamma}^{*}u_i \rangle &\leq \varepsilon_n, \\ \sum_{i=1}^{k} \alpha_{n,i}(c_i - T_{\gamma}^{*}u_i) &\geq -\varepsilon_n, \text{ on } \mathcal{X} \times \mathcal{A} \\ \alpha_n \geq 0, \ \sum_{i=1}^{k} \alpha_{n,i} = 1. \end{cases} \tag{18}$$

Since the sequence $\{\alpha_n\}_n$ is bounded in $\mathbb{R}^k$, there exists a subsequence $\{\alpha_{n_l}\}_{l=1}^{\infty}$ such that $\lim_{l \to \infty} \alpha_{n_l} = \alpha$, for some $\alpha \in \mathbb{R}^k$. Taking the $l \to \infty$ in (18), we get (17) and so $c = \sum_{i=1}^{k} \alpha_i c_i \in \mathcal{C}(\pi_{\mathrm{E}})$. This concludes the proof. $\qquad \square$

## B.4 Proof of Proposition 3.3

*Proof.* It is easy to check that for every $\tilde{c} \in \mathcal{C}^{\varepsilon}(\pi_{\mathrm{E}})$, there exist $c \in \mathcal{C}(\pi_{\mathrm{E}})$ such that $\langle \mu_{\nu_0}^{\pi_{\mathrm{E}}}, \tilde{c} - c \rangle \leq \varepsilon$, and $c - \tilde{c} \leq \varepsilon$, on $\mathcal{X} \times \mathcal{A}$. For example, if $\tilde{u} \in \mathrm{Lip}(\mathcal{X})$ such that

$$\begin{cases} \langle \mu_{\nu_0}^{\pi_{\mathrm{E}}}, \tilde{c} - T_{\gamma}^{*}\tilde{u} \rangle &\leq \varepsilon, \\ \tilde{c} - T_{\gamma}^{*}\tilde{u} &\geq -\varepsilon, \text{ on } \mathcal{X} \times \mathcal{A}, \end{cases} \tag{19}$$

then we may choose $c = T_{\gamma}^{*}\tilde{u}$. Let $\tilde{\pi}$ be an optimal policy for (MDP$_{\tilde{c}}$). Then,

$$V_c^{\tilde{\pi}}(\nu_0) - V_c^{\pi_{\mathrm{E}}}(\nu_0) = \langle \mu_{\nu_0}^{\tilde{\pi}} - \mu_{\nu_0}^{\pi_{\mathrm{E}}}, c \rangle$$

$$= \left\langle \mu_{\nu_0}^{\tilde{\pi}}, c - \tilde{c} \right\rangle + \underbrace{\left\langle \mu_{\nu_0}^{\tilde{\pi}} - \mu_{\nu_0}^{\pi_{\mathrm{E}}}, \tilde{c} \right\rangle}_{\leq 0} + \left\langle \mu_{\nu_0}^{\pi_{\mathrm{E}}}, \tilde{c} - c \right\rangle$$

$$\leq \left\langle \mu_{\nu_0}^{\tilde{\pi}} - \mu_{\nu_0}^{\pi_{\mathrm{E}}}, c - \tilde{c} \right\rangle$$

$$\leq \frac{2 - \gamma}{1 - \gamma} \varepsilon$$

$\square$

## B.5 Proof of Proposition 4.1

Note that Proposition 4.1 is a minor extension from [48, Prop. 7].

*Proof.* Let $c \in \mathcal{C}(\pi_{\mathrm{E}})$, then there exists a certificate $u \in \mathrm{Lip}(\mathcal{X})$ such that $c - T_\gamma^* u \geq 0$, on $\mathcal{X} \times \mathcal{A}$ and $\left\langle \mu_{\nu_0}^{\pi_{\mathrm{E}}}, c - T_\gamma^* u \right\rangle = 0$. By Theorem 3.1 and its proof, we get that $c$ is inverse feasible and $u = V_c^\star$ $\nu_0$-almost everywhere (a.e.). Thus, if $c \equiv C$, for some constant $C$, then $u \equiv \frac{C}{1-\gamma}$, $\nu_0$-a.e. We then have $c - T_\gamma^* u = 0$, $\nu_0$-a.e., and so $\int_{\mathcal{X} \times \mathcal{A}} (c - T_\gamma^* u) \mathrm{d}x \mathrm{d}a = 0$. $\square$

## B.6 Proof of Proposition 4.2

For $p \in [1, \infty]$, we denote by $\|\cdot\|_p$ the $p$-norm in $\mathbb{R}^n$ and by $\boldsymbol{x} \cdot \boldsymbol{y}$ the usual inner product.

*Proof.* We consider the following tightening of the semi-infinite convex program (IP).

$$J_{n_{c,u}} \triangleq \begin{cases} \inf_{\alpha,\beta,\varepsilon} & \varepsilon \\ \text{s.t.} & \left\langle \mu_{\nu_0}^{\pi_{\mathrm{E}}}, c - T_\gamma^* u \right\rangle \leq \varepsilon, \\ & c - T_\gamma^* u \geq 0, \text{ on } \mathcal{X} \times \mathcal{A}, \\ & \int_{\mathcal{X} \times \mathcal{A}} (c(x,a) - T_\gamma^* u(x,a)) \, \mathrm{d}(x,a) = 1, \\ & c \in \mathbf{C}_{n_c}, u \in \mathbf{U}_{n_u}, \varepsilon \geq 0 \end{cases}$$

$$= \begin{cases} \inf_{\alpha,\beta} & \left\langle \mu_{\nu_0}^{\pi_{\mathrm{E}}}, c \right\rangle - \left\langle \nu_0, u \right\rangle \\ \text{s.t.} & c - T_\gamma^* u \geq 0, \text{ on } \mathcal{X} \times \mathcal{A}, \\ & \int_{\mathcal{X} \times \mathcal{A}} (c(x,a) - T_\gamma^* u(x,a)) \, \mathrm{d}(x,a) = 1, \\ & c \in \mathbf{C}_{n_c}, u \in \mathbf{U}_{n_u}, \end{cases} \tag{20}$$

where the last equality follows by using that $\left\langle \mu_{\nu_0}^{\pi_{\mathrm{E}}}, T_\gamma^* u \right\rangle = \left\langle T_\gamma \mu_{\nu_0}^{\pi_{\mathrm{E}}}, u \right\rangle = \left\langle \nu_0, u \right\rangle$ and an epigraphic transformation.

The assumption that $u_1 \equiv 1$ and $\theta > \frac{1}{(1-\gamma)\mathrm{leb}(\mathcal{X} \times \mathcal{A})}$ ensures feasibility of the convex program (20) and by Assumption 2.1 (A1) the feasibility set is compact and thus the optimal value is finite and is attained. Note that $\mathrm{leb}(\mathcal{X} \times \mathcal{A})$ denotes the Lebesgue measure of $\mathcal{X} \times \mathcal{A}$. Moreover, since (20) is a tightening of (IP) it holds that $\tilde{\varepsilon} \leq J_{n_{c,u}}$.

Consider the infinite-dimensional version of (20),

$$J \triangleq \begin{cases} \inf_{c,u} & \left\langle \mu_{\nu_0}^{\pi_{\mathrm{E}}}, c \right\rangle - \left\langle \nu_0, u \right\rangle \\ \text{s.t.} & c - T_\gamma^* u \geq 0, \text{ on } \mathcal{X} \times \mathcal{A}, \\ & \int_{\mathcal{X} \times \mathcal{A}} (c(x,a) - T_\gamma^* u(x,a)) \, \mathrm{d}(x,a) = 1, \\ & c \in \mathrm{Lip}(\mathcal{X} \times \mathcal{A}), u \in \mathrm{Lip}(\mathcal{X}). \end{cases} \tag{21}$$

By the characterization of the inverse feasibility set, we have that $J = 0$ and $(c^\star, u^\star)$ is an optimal solution for (21) [4]. Note that (21) can be expressed in the standard conic form

$$J = \begin{cases} \inf_{\boldsymbol{x}} & \left\langle \boldsymbol{l_0}, \boldsymbol{x} \right\rangle \\ \text{s.t.} & \boldsymbol{Ax} - \boldsymbol{b_0} \in \boldsymbol{K}, \\ & \boldsymbol{x} \in \boldsymbol{X}, \end{cases} \tag{22}$$

where

---

[4]Without loss of generality we may assume that $\int_{\mathcal{X} \times \mathcal{A}} (c^\star - T_\gamma^* u^\star)(x,a) \, \mathrm{d}(x,a)$ since we can always rescale the optimal cost and value function by the same scale factor.

- $(\boldsymbol{X}, \boldsymbol{L})$ is a dual pair of vector spaces and $\boldsymbol{l_0} \in \boldsymbol{L}$;

- $(\boldsymbol{B}, \boldsymbol{Y})$ is a dual pair of vector spaces and $\boldsymbol{b_0} \in \boldsymbol{B}$;

- $\boldsymbol{K}$ is a positive cone in $\boldsymbol{B}$ and $\boldsymbol{K}^*$ is its dual cone in $\boldsymbol{Y}$, i.e.,

$$\boldsymbol{K}^* = \{\boldsymbol{y} \in \boldsymbol{Y} : \langle \boldsymbol{y}, \boldsymbol{b} \rangle \geq 0, \ \forall \boldsymbol{b} \in \boldsymbol{B}\};$$

- $\boldsymbol{\mathcal{A}} : \boldsymbol{X} \to \boldsymbol{B}$ is linear and continuous with respect to the induced weak topologies.

Indeed, this is the case if we introduce the following:

$$\boldsymbol{X} \triangleq \mathrm{Lip}(\mathcal{X} \times \mathcal{A}) \times \mathrm{Lip}(\mathcal{X}), \qquad \boldsymbol{L} \triangleq \mathcal{M}(\mathcal{X} \times \mathcal{A}) \times \mathcal{M}(\mathcal{X}),$$
$$\boldsymbol{B} \triangleq \mathrm{Lip}(\mathcal{X} \times \mathcal{A}) \times \mathbb{R}, \qquad \boldsymbol{Y} \triangleq \mathcal{M}(\mathcal{X} \times \mathcal{A}) \times \mathbb{R},$$
$$\boldsymbol{K} \triangleq \mathrm{Lip}(\mathcal{X} \times \mathcal{A})_+ \times \{0\}, \qquad \boldsymbol{K^*} \triangleq \mathcal{M}(\mathcal{X} \times \mathcal{A})_+ \times \mathbb{R},$$
$$\boldsymbol{b_0} \triangleq (\boldsymbol{0}, 1), \qquad \boldsymbol{l_0} \triangleq (\mu_{\nu_0}^{\pi_E}, -\nu_0),$$
$$\boldsymbol{\mathcal{A}}(c, u) \triangleq \begin{bmatrix} \boldsymbol{\mathcal{A}_1}(c, u) \\ \boldsymbol{\mathcal{A}_2}(c, u) \end{bmatrix} \triangleq \begin{bmatrix} c - T_\gamma^* u \\ \int_{\mathcal{X} \times \mathcal{A}} (c - T_\gamma^* u)(x, a) \ \mathrm{d}(x, a) \end{bmatrix}.$$

On the pair $(\boldsymbol{X}, \boldsymbol{C})$ we consider the norms

$$\|\boldsymbol{x}\| = \|(c, u)\| \triangleq \max\{\|c\|_{\mathrm{L}}, \|u\|_{\mathrm{L}}\}, \ \boldsymbol{x} = (c, u) \in \boldsymbol{X},$$

$$\begin{aligned} \|\boldsymbol{l}\|_* &= \sup_{\|\boldsymbol{x}\| \leq 1} \langle \boldsymbol{l}, \boldsymbol{x} \rangle = \sup_{\|c\|_{\mathrm{L}} \leq 1} \langle l_1, c \rangle + \sup_{\|u\|_{\mathrm{L}} \leq 1} \langle l_2, u \rangle \\ &= \|l_1\|_{\mathrm{W}} + \|l_2\|_{\mathrm{W}}, \ \boldsymbol{l} = (l_1, l_2) \in \boldsymbol{L}, \end{aligned}$$

which are dual to each other. Similarly, on the pair $(\boldsymbol{B}, \boldsymbol{Y})$ we consider the norms

$$\begin{aligned} \|(b_1, b_2)\| &= \max\{\|b_1\|_{\mathrm{L}}, |b_2|\}, \\ \|(y_1, y_2)\|_* &= \|y_1\|_{\mathrm{W}} + |y_2|. \end{aligned}$$

With this notation in mind, by virtue of [31, Th. 3.3] we have that

$$\begin{aligned} \tilde{\varepsilon} &\leq J_{n_{c,u}} - \underbrace{J}_{=0} \\ &\leq (\|\boldsymbol{l_0}\|_* + \mathcal{D}_{\gamma, \theta} \|\boldsymbol{\mathcal{A}}\|_{\mathrm{op}}) \\ &\qquad (\|c^\star - \Pi_{\mathbf{C}_{n_c}}(c^\star)\|_{\mathrm{L}} + \|u^\star - \Pi_{\mathbf{U}_{n_u}}(u^\star)\|_{\mathrm{L}}), \end{aligned} \tag{23}$$

where $\mathcal{D}_{\gamma, \theta}$ is an upper bound of a dual optimizer of (20) with respect to an appropriately defined dual norm, and $\|\boldsymbol{\mathcal{A}}\|_{\mathrm{op}}$ is the operator norm of $\boldsymbol{\mathcal{A}}$. We will next compute the involved quantities in (23).

We have

$$\|\boldsymbol{l_0}\|_* = \|\mu_{\nu_0}^{\pi_E}\|_{\mathrm{W}} + \|\nu_0\|_{\mathrm{W}} = \frac{1}{1 - \gamma} + 1. \tag{24}$$

Next note that

$$\begin{aligned} \|\mathsf{P}u\|_{\mathrm{L}} &= \|\mathsf{P}u\|_\infty + |\mathsf{P}u|_{\mathrm{L}} \leq \|u\|_\infty + L_{\mathsf{P}} |u|_{\mathrm{L}} \\ &\leq \max\{1, L_{\mathsf{P}}\} \|u\|_{\mathrm{L}}, \end{aligned}$$

where in the first inequality we used that $\mathsf{P}$ is a stochastic kernel and Assumption 2.1 (A2). Therefore,

$$\begin{aligned} \|\boldsymbol{\mathcal{A}_1}(c, u)\|_{\mathrm{L}} &= \|c - u + \mathsf{P}u\|_{\mathrm{L}} \\ &\leq (2 + \gamma \max\{1, L_{\mathsf{P}}\}) \|(c, u)\|. \end{aligned}$$

Moreover,

$$\begin{aligned} |\boldsymbol{\mathcal{A}_2}(c, u)| &\leq (\|c\|_\infty + (1 + \gamma) \|u\|_\infty) \dim(\mathcal{X} \times \mathcal{A}) \\ &\leq (2 + \gamma) \mathrm{leb}(\mathcal{X} \times \mathcal{A}) \|(c, u)\|. \end{aligned}$$

All in all,

$$\|\boldsymbol{\mathcal{A}}\|_{\mathrm{op}} \triangleq \sup_{\|(c, u)\| \leq 1} \|\boldsymbol{\mathcal{A}}(c, u)\|$$

$$\leq \quad \max\{(2+\gamma)\mathrm{leb}(\mathcal{X}\times\mathcal{A}), 2+\gamma\max\{1, L_{\mathsf{P}}\}\}$$
$$\leq \quad (2+\gamma)\max\{1, L_{\mathsf{P}}, \mathrm{leb}(\mathcal{X}\times\mathcal{A})\}. \tag{25}$$

It remains to compute the constant $\mathcal{D}_{\gamma,\theta}$. To this aim, let $\{\boldsymbol{x}_i\}_{i=1}^{n_c+n_u}$ be basis elements in $\boldsymbol{X}$ given by $\boldsymbol{x}_i = (c_i, 0)$, for $i = 1, \dots, n_c$ and $\boldsymbol{x}_i = (0, u_i)$, for $i = n_c+1, \dots, n_c+n_u$. These are linearly independent by assumption. Then, we define the linear operator $\boldsymbol{\mathcal{A}}_{n_{c,u}} : \mathbb{R}^{n_c+n_u} \to \boldsymbol{Y}$ by $\boldsymbol{\mathcal{A}}_{n_{c,u}}(\boldsymbol{\rho}) = \sum_{i=1}^{n_c+n_u}\rho_i\boldsymbol{\mathcal{A}}\boldsymbol{x_i} = \sum_{i=1}^{n_c}\alpha_i\boldsymbol{\mathcal{A}}\boldsymbol{x_i} + \sum_{i=n_c+1}^{n_c+n_u}\beta_i\boldsymbol{\mathcal{A}}\boldsymbol{x_i}$, for $\boldsymbol{\rho} = (\alpha, \beta) \in \mathbb{R}^{n_c+n_u}$. Then, it is easy to see that its adjoint $\boldsymbol{\mathcal{A}}_{n_{c,u}}^* : \boldsymbol{Y} \to \mathbb{R}^{n_c+n_u}$ is given by $\boldsymbol{\mathcal{A}}_{n_{c,u}}^*(\boldsymbol{y}) = [\langle\boldsymbol{\mathcal{A}}\boldsymbol{x_1}, \boldsymbol{y}\rangle, \dots, \langle\boldsymbol{\mathcal{A}}\boldsymbol{x_{n_c+n_u}}, \boldsymbol{y}\rangle]$. On $\mathbb{R}^{n_c+n_u}$ we consider the norm
$$\|\boldsymbol{\rho}\|_{\mathcal{R}} = \|(\alpha, \beta)\|_{\mathcal{R}} \triangleq \max\{\|\alpha\|_1, \|\beta\|_1\}$$

Moreover, we set
$$\tilde{\boldsymbol{l}_0} \triangleq [\underbrace{\langle\mu_{\nu_0}^{\pi_{\mathrm{E}}}, c_1\rangle, \dots, \langle\mu_{\nu_0}^{\pi_{\mathrm{E}}}, c_{n_c}\rangle}_{=\tilde{l}_{0,1}}, \underbrace{\langle-\nu_0, u_1\rangle, \dots, \langle-\nu_0, u_{n_u}\rangle}_{=\tilde{l}_{0,2}}]$$

Then, the semi-infinite convex program (20) can be written in the form
$$J_{n_{c,u}} = \begin{cases} \inf_{\boldsymbol{\rho}} & \tilde{\boldsymbol{l}_0} \cdot \boldsymbol{\rho} \\ \text{s.t.} & \boldsymbol{\mathcal{A}}_{n_{c,u}}\boldsymbol{\rho} - \boldsymbol{b_0} \in \boldsymbol{K}, \\ & \|\boldsymbol{\rho}\|_{\mathcal{R}} \leq \theta, \ \boldsymbol{\rho} \in \mathbb{R}^{n_c+n_u}. \end{cases} \tag{26}$$

Dualizing the conic inequality constraint in (26) and using the dual norm definition, we get its dual
$$\tilde{J}_{n_{c,u}} = \begin{cases} \sup_{\boldsymbol{y}\in\boldsymbol{Y}} & \langle\boldsymbol{b_0}, \boldsymbol{y}\rangle - \theta\|\boldsymbol{\mathcal{A}}_{n_{c,u}}^*\boldsymbol{y} - \tilde{\boldsymbol{l}_0}\|_{\mathcal{R}^*} \\ \text{s.t.} & \boldsymbol{y} \in \boldsymbol{K^*}. \end{cases} \tag{27}$$

Let $\boldsymbol{y}^\star$ be a dual optimizer for (26). Assume that there exists a constant $C > 0$ such that
$$\|\boldsymbol{\mathcal{A}}_{n_{c,u}}^*\boldsymbol{y}^\star\|_{\mathcal{R}^*} \geq C\|\boldsymbol{y}^\star\|_*.$$
Then by virtue of [31, Prop. 3.2], we have the bound
$$\|\boldsymbol{y}^\star\|_* \leq \frac{2\theta\|\tilde{\boldsymbol{l}_0}\|_{R^*}}{C\theta - \|\boldsymbol{b_0}\|} \leq \mathcal{D}_{\gamma,\theta}. \tag{28}$$
To compute the constant $\mathcal{D}_{\gamma,\theta}$, we need to bound the involved quantities in (28).

We will first show that $y_2^\star \geq 0$. By Sion's minimax Theorem [75] the duality gap between (26) and (27) is zero, i.e., $J_{n_{c,u}} = \tilde{J}_{n_{c,u}}$. Note however that by construction $J_{n_{c,u}} \geq 0$, since for any feasible $\boldsymbol{\rho}$ to (26) it holds that $\tilde{\boldsymbol{l}_0} \cdot \boldsymbol{\rho} = \langle\mu_{\nu_0}^{\pi_{\mathrm{E}}}, \boldsymbol{\mathcal{A}}_{n_{c,u}}\boldsymbol{\rho} - \boldsymbol{b_0}\rangle \geq 0$. Then,
$$0 \leq J_{n_{c,u}} = \tilde{J}_{n_{c,u}} = \langle\boldsymbol{b_0}, \boldsymbol{y}^\star\rangle - \theta\|\boldsymbol{\mathcal{A}}_{n_{c,u}}^*\boldsymbol{y}^\star - \tilde{\boldsymbol{l}_0}\|_{\mathcal{R}^*}$$
$$= y_2^\star - \theta\|\boldsymbol{\mathcal{A}}_{n_{c,u}}^*\boldsymbol{y}^\star - \tilde{\boldsymbol{l}_0}\|_{\mathcal{R}^*}.$$

Thus, $y_2^\star \geq 0$. Therefore,
$$\|\boldsymbol{\mathcal{A}}_{n_{c,u}}^*\boldsymbol{y}^\star\|_{\mathcal{R}^*} \quad \geq \quad |\langle\boldsymbol{\mathcal{A}}\boldsymbol{x_{n_c+1}}, \boldsymbol{y}^\star\rangle| = |\langle\boldsymbol{A}(0, u_1), \boldsymbol{y}^\star\rangle|$$
$$= \quad |\langle T_\gamma^* u_1, y_1^\star\rangle + y_2^\star \int_{\mathcal{X}\times\mathcal{A}} T_\gamma^* u_1 \ \mathrm{d}(x, a)|$$
$$= \quad (1-\gamma)\|y_1^\star\|_{\mathsf{W}} + (1-\gamma)\mathrm{dim}(\mathcal{X}\times\mathcal{A})|y_2^\star|$$
$$\geq \quad \underbrace{(1-\gamma)\min\{1, \mathrm{leb}(\mathcal{X}\times\mathcal{A})\}}_{\triangleq C}\|\boldsymbol{y}^\star\|_*,$$

where we used that $u_1 \equiv 1$, $y_1^\star \in \mathcal{M}(\mathcal{X}\times\mathcal{A})_+$ and so $\|y_1^\star\|_{\mathsf{W}} = y_1^\star(\mathcal{X}\times\mathcal{A})$, and $y_2^\star \geq 0$.

In addition, a direct computation gives,
$$\|\tilde{\boldsymbol{l}_0}\|_{R^*} = \sup_{\|\boldsymbol{\rho}\|_{\mathcal{R}}\leq 1} \tilde{\boldsymbol{l}_0} \cdot \boldsymbol{\rho} = \|\tilde{l}_{0,1}\|_\infty + \|\tilde{l}_{0,2}\|_\infty \leq \frac{K_{c,\infty}}{1-\gamma} + K_{u,\infty}.$$

Putting them all together in (28), we get
$$\|\boldsymbol{y}^\star\|_* \leq \frac{2\theta\|\tilde{\boldsymbol{l}_0}\|_{R^*}}{C\theta - \|\boldsymbol{b_0}\|} \leq \frac{2\theta(K_{c,\infty} + K_{u,\infty})}{(1-\gamma)^2\min\{1, d\}\theta + \gamma - 1} \triangleq \mathcal{D}_{\gamma,\theta}, \tag{29}$$
where we used that $\|\boldsymbol{b_0}\| = 1$. A combination of (23), (24), (25) and (29) ends the proof. $\qquad\square$

## B.7 Proof of Theorem 4.1

The symbol $\models$ refers to feasibility satisfaction, i.e., $x \models \mathcal{R}$ means that $x$ is a feasible solution for the program $\mathcal{R}$.

*Proof.* The proof is a consequence of [29, Theorem 1], [30, Lemma 3.2] and [30, Proposition 3.2]. Let us denote the optimization variable of ($\text{SIP}_\mathbf{N}$) by $z = (\alpha, \beta, \varepsilon) \in \mathbb{R}^{n_c + n_u + 1}$, where $\alpha = (\alpha_1, \ldots, \alpha_{n_c})$ and $\beta = (\beta_1, \ldots, \beta_{n_u})$. Let $\lambda = (x, a)$ be the uncertainty parameter and $\Lambda = \mathcal{X} \times \mathcal{A}$ the uncertainty set. Consider the function

$$f(z, \lambda) = -\sum_{i=1}^{n_c} \alpha_i c_i(x, a) + \sum_{j=1}^{n_u} \beta_j(u_j(x) - \gamma \mathsf{P} u_j(x, a)) - \varepsilon$$

and the set

$$\mathcal{Z} = \left\{ z = (\alpha, \beta, \varepsilon) \in \mathbb{R}^{n_c + n_u + 1} : \|\alpha\|_2 \leq \theta, \|\beta\|_2 \leq \theta, \right.$$
$$\left. \langle \mu_{\nu_0}^{\pi_\mathrm{E}}, -f(z, \cdot) - \varepsilon \rangle \leq \varepsilon, \int_\Lambda (-f(z, \lambda) + \varepsilon) d\lambda = 1, \varepsilon \geq 0 \right\}.$$

Note that $\mathcal{Z} \subset \mathbb{R}^{n_c + n_u + 1}$ is convex and compact and independent of $\lambda \in \Lambda$. We show that the set $\mathcal{Z}$ is nonempty. As noted in the proof of Proposition 4.2 (Appendix B.6) the assumption that $u_1 \equiv 1$ and $\theta > \frac{1}{(1-\gamma)\mathrm{leb}(\mathcal{X} \times \mathcal{A})}$ considered in Proposition 4.2 and Theorems 4.1-4.2 ensures the feasibility of the convex program (IP) which in particular implies that $\mathcal{Z} \neq \emptyset$. Here $\mathrm{leb}(\mathcal{X} \times \mathcal{A})$ denotes the Lebesgue measure of $\mathcal{X} \times \mathcal{A}$.

Note that the function $f : \mathcal{Z} \times \Lambda \to \mathbb{R}$ is measurable, and linear in the first argument for all $\lambda \in \Lambda$. Moreover, by Assumption 2.1 (A1)-(A2), we get that $f$ is bounded in the second argument for all $z \in \mathcal{Z}$, and $f$ is Lipschitz continuous on $\Lambda$ uniformly in $\mathcal{Z}$ with Lipschiz constant

$$L_\Lambda \triangleq \theta \sqrt{n_c} L_c + \theta \sqrt{n_u} (L_u L_\mathsf{P} + L_u).$$

After introducing this notation, one can see that the random convex program ($\text{SIP}_\mathbf{N}$) can be written as,

$$\text{SIP}_\mathbf{N} : \begin{cases} \inf_{z \in \mathcal{Z}} & h^\top z \\ \text{s.t.} & f(z, \lambda^{(\ell)}) \leq 0, \forall \ell = 1, \ldots, N, \end{cases}$$

where $h = (\mathbf{0}_{\mathbb{R}^{n_c}}, \mathbf{0}_{\mathbb{R}^{n_u}}, 1)$ and the multisample $\{\lambda^{(i)} = (x^{(\ell)}, a^{(\ell)})\}_{\ell=1}^N$ is a random element on the product probability space $(\Lambda^N, \mathcal{B}(\Lambda)^N, \mathbb{P}^N)$. Moreover, ($\text{SIP}_\mathbf{N}$) is the scenario counterpart of (IP),

$$\text{IP} : \begin{cases} \inf_{z \in \mathcal{Z}} & h^\top z \\ \text{s.t.} & f(z, \lambda) \leq 0, \forall \lambda \in \Lambda, \end{cases}$$

where one enforces constraint satisfaction for all the realizations of the uncertainty. Clearly if $z = (\alpha, \beta, \varepsilon) \models \text{IP}$, then the associated cost function $c = \sum_{i=1}^{n_c} \alpha_i c_i \in \mathcal{C}^\varepsilon(\pi_\mathrm{E})$.

For a fixed reliability level $\epsilon \in (0, 1)$, the associated chance-constrained program is given by

$$\text{CCP}_\epsilon : \begin{cases} \inf_{z \in \mathcal{Z}} & h^\top z \\ \text{s.t.} & \mathbb{P}[\lambda \in \Lambda : f(z, \lambda) \leq 0] \geq 1 - \epsilon, \end{cases}$$

where one allows constraint violation with low probability. Now the assumption that $u_1 \equiv 1$ and $\theta > \frac{1}{(1-\gamma)\dim(\mathcal{X} \times \mathcal{A})}$ is sufficient for the existence of a Slater point for the robust convex program (IP), i.e., there exists $z_0 \in \mathcal{Z}$ such that $\sup_{\lambda \in \Lambda} f(z_0, \lambda) < 0$. By this fact and by Assumption 4.2, we can apply [29, Theorem 1] and conclude that for $N \geq \mathrm{N}(n_c + n_u + 1, \epsilon, \delta)$ it holds that

$$\mathbb{P}^N[\tilde{z}_N \models \text{CCP}_\epsilon] \geq 1 - \delta, \tag{30}$$

where $\tilde{z}_N = (\tilde{\alpha}_N, \tilde{\beta}_N, \tilde{\varepsilon}_N)$ is the optimizer of ($\text{SIP}_\mathbf{N}$). Note that by Assumption 4.2, the optimizer $\tilde{z}_N$ is uniquely defined and is a $\mathcal{Z}$-valued random variable on $(\Lambda^N, \mathcal{B}(\Lambda)^N, \mathbb{P}^N)$.

Consider now the $\zeta$-perturbed robust convex program for some $\zeta > 0$,

$$\text{IP}_\zeta : \begin{cases} \inf_{z \in \mathcal{Z}} & h^\top z \\ \text{s.t.} & f(z, \lambda) \leq \zeta, \ \forall \lambda \in \Lambda. \end{cases}$$

Note that due to the min-max structure of (IP), the mapping from $\zeta$ to the optimal value of $\text{IP}_\zeta$ is Lipschitz continuous with Lipschitz constant 1 [30, Remark 3.5].

We have the following implication,

$$z = (\alpha, \beta, \varepsilon) \models \text{IP}_\zeta \Rightarrow c = \sum_{i=1}^{n_c} \alpha_i c_i \in \mathcal{C}^{\varepsilon + \zeta}(\pi_\text{E}). \tag{31}$$

Moreover, under Assumption 4.1 and since $f(z, \cdot)$ is Lipschitz continuous on $\Lambda$ uniformly in $\mathcal{Z}$ with Lipschitz constant $L_\Lambda$, by virtue of [30, Lemma 3.2] and [30, Proposition 3.8], we have that

$$z \models \text{CCP}_\epsilon \Rightarrow z \models \text{IP}_{\mathbf{h}(\epsilon)}, \tag{32}$$

where

$$h(\epsilon) \triangleq L_\Lambda g^{-1}(\epsilon). \tag{33}$$

Combining (30), (31), (32) and (33) we get that for $N \geq \text{N}(n_c + n_u + 1, g(\frac{\epsilon}{L_\Lambda}), \delta)$,

$$\mathbb{P}^N[\tilde{c}_N \in \mathcal{C}^{\tilde{\varepsilon} + \epsilon}(\pi_\text{E})] \geq 1 - \delta.$$

Finally notice that since $(\text{SIP}_\mathbf{N})$ is a relaxation of (IP) we have $\tilde{\varepsilon} \leq \varepsilon_\text{approx}$ with probability 1. $\qquad\square$

### B.8 Proof of Theorem 4.2

For the remaining analysis we will use for brevity the following notation,

$$\mathbb{P}_\zeta \triangleq \mathbb{P}^N, \quad \mathbb{P}_\tau \triangleq (\mathbb{P}_{\nu_0}^{\pi_\text{E}})^m, \quad \mathbb{P}_\xi \triangleq \nu_0{}^n$$

and adopt a similar notation for products of probability measures, e.g., $\mathbb{P}_{\tau, \xi} \triangleq \mathbb{P}_\tau \otimes \mathbb{P}_\xi$. We first need the following result.

**Proposition B.1.** *Let $\epsilon \in (0, 1)$ and $\delta \in (0, 1)$. Under Assumption 2.1and (A1), for $n \geq \frac{8 K_{u,\infty}^2 \theta^2 n_u \ln(\frac{8 n_u}{\delta})}{\epsilon^2}$ and $m \geq \frac{8 K_{c,\infty}^2 \theta^2 n_c \ln(\frac{8 n_c}{\delta})}{(1-\gamma)^2 \epsilon^2}$, it holds with probability at least $1 - \delta/2$ that*

$$\sup_{c \in \mathbf{C}_{n_c}, u \in \mathbf{U}_{n_u}} \left| \langle \mu_{\nu_0}^{\pi_\text{E}}, c - T_\gamma^* u \rangle - \widehat{\langle \mu_{\nu_0}^{\pi_\text{E}}, c \rangle} + \widehat{\langle \nu_0, u \rangle} \right| \leq \epsilon,$$

*where $K_{c,\infty} \triangleq \max_{i=1,\ldots,n_c} \|c_i\|_\infty$ and $K_{u,\infty} \triangleq \max_{j=1,\ldots,n_u} \|u_j\|_\infty$.*

*Proof.* First note that under Assumption (A1), the quantities $K_{c,\infty}$ and $K_{u,\infty}$ are finite. Now by using the Hoeffding's bound, we have that for $n \geq \frac{8 K_{u,\infty}^2 \theta^2 n_u \ln(\frac{8 n_u}{\delta})}{\epsilon^2}$,

$$\mathbb{P}_\xi \left[ \left| \langle \nu_0, u_j \rangle - \widehat{\langle \nu_0, u_j \rangle} \right| \leq \frac{\epsilon}{2 \sqrt{n_u} \theta} \right] \geq 1 - \frac{\delta}{4 n_u}, \tag{34}$$

for all $j = 1, \ldots, n_u$. Therefore,

$$\mathbb{P}_\xi \left[ \sup_{u \in \mathbf{U}_{n_u}} \left| \langle \nu_0, u \rangle - \widehat{\langle \nu_0, u \rangle} \right| \leq \frac{\epsilon}{2} \right]$$
$$\geq \mathbb{P}_\xi \left[ \forall j = 1, \ldots, n_u : \left| \langle \nu_0, u_j \rangle - \widehat{\langle \nu_0, u_j \rangle} \right| \leq \frac{\epsilon}{2 \sqrt{n_u} \theta} \right]$$
$$\geq 1 - \delta/4,$$

where the first inequality follows by the monotonicity property of probability measures and the second one follows by (34) and a union bound. Integrating over the whole $(\Omega^m, \mathbb{P}_\tau)$, we end up that

$$\mathbb{P}_{\tau, \xi} [\underbrace{\sup_{u \in \mathbf{U}_{n_u}} \left| \langle \nu_0, u \rangle - \widehat{\langle \nu_0, u \rangle} \right| \leq \frac{\epsilon}{2}}_{\triangleq A}] \geq 1 - \delta/4. \tag{35}$$

By using analogous arguments and taking into account that $|\sum_{t=0}^{\infty} \gamma^t c_i(x_t, a_t)| \leq \frac{K_{c,\infty}}{1-\gamma}$, for all $(x_t, a_t) \in \mathcal{X} \times \mathcal{A}$, $t \in \mathbf{N}$, $i = 1, \ldots, n_c$, we can conclude that for $m \geq \frac{8K_{c,\infty}^2 \theta^2 n_c \ln\left(\frac{8n_c}{\delta}\right)}{(1-\gamma)^2 \epsilon^2}$,

$$\mathbb{P}_{\tau,\xi}[\underbrace{\sup_{c \in \mathbf{C}_{n_c}} \left|\langle \mu_{\nu_0}^{\pi_\mathrm{E}}, c \rangle - \widehat{\langle \mu_{\nu_0}^{\pi_\mathrm{E}}, c \rangle}\right| \leq \frac{\epsilon}{2}}_{\triangleq B}] \geq 1 - \delta/4. \tag{36}$$

Finally, note that

$$\mathbb{P}_{\tau,\xi}\left[\sup_{c \in \mathbf{C}_{n_c}, u \in \mathbf{U}_{n_u}} \left|\langle \mu_{\nu_0}^{\pi_\mathrm{E}}, c - T_\gamma^* u \rangle - \widehat{\langle \mu_{\nu_0}^{\pi_\mathrm{E}}, c \rangle} + \widehat{\langle \nu_0, u \rangle}\right| \leq \epsilon\right]$$
$$\geq \mathbb{P}_{\tau,\xi}[A \cap B]$$
$$\geq 1 - \delta/2,$$

where in the first inequality we have used the monotonicity property of probability measures and the second inequality follows by (35), (36) and a simple union bound. $\qquad\square$

*Proof of Theorem 4.2.* By using the same notation as in the proof of Theorem 4.1, we can write

$$\mathsf{SIP_{N,m,n}} : \begin{cases} \inf_{z \in \mathcal{Z}_{m,n}} & h^\top z \\ \text{s.t.} & f(z, \lambda^{(i)}) \leq 0, \forall i = 1, \ldots, N, \end{cases}$$

where

$$\mathcal{Z}_{m,n} = \big\{ z = (\alpha, \beta, \varepsilon) \in \mathbb{R}^{n_c+n_u+1} : \|\alpha\|_2 \leq \theta, \|\beta\|_2 \leq \theta,$$

$$\sum_{i=1}^{n_c} \alpha_i \widehat{\langle \mu_{\nu_0}^{\pi_\mathrm{E}}, c_i \rangle}(\tau) - \sum_{j=1}^{n_u} \beta_j \widehat{\langle \nu_0, u_j \rangle}(\xi) \leq \varepsilon,$$

$$\int_\Lambda (-f(z, \lambda) + \varepsilon)d\lambda = 1, \varepsilon \geq 0 \big\}.$$

Let $\tilde{z}_{N,m,n} = (\tilde{\alpha}_{N,m,n}, \tilde{\beta}_{N,m,n}, \tilde{\varepsilon}_{N,m,n})$ be the optimizer of $\mathsf{SIP_{N,m,n}}$ and let $\tilde{c}_{N,m,n} = \sum_{i=1}^{n_c} \tilde{\alpha}_{N,m,n_i} c_i$ be the associated cost function.

We first fix multi-samples $\tau, \xi$. Similarly as in Theorem 4.1, we can conclude that for $N \geq \mathrm{N}(n_c + n_u + 1, g(\frac{\epsilon}{L_\Lambda}), \delta/2)$,

$$\mathbb{P}_\zeta[y \in \Lambda^N : \tilde{z}_{N,m,n}(y, \tau, \xi) \models \mathsf{IP_{m,n,\epsilon}}(y, \tau)] \geq 1 - \delta/2,$$

where $\mathsf{IP_{m,n,\epsilon}}$ is the $\epsilon$-perturbed robust counterpart of $\mathsf{SIP_{N,m,n}}$ given by

$$\mathsf{IP_{m,n,\epsilon}} : \begin{cases} \inf_{z \in \mathcal{Z}_{m,n}} & h^\top z \\ \text{s.t.} & f(z, \lambda) \leq \epsilon, \forall \lambda \in \Lambda, \end{cases}$$

Integrating over the whole probability space $(\Omega^m \otimes \mathcal{X}^n, \mathbb{P}_{\tau,\xi})$, we get

$$\mathbb{P}_{y,\tau,\xi}[\tilde{z}_{N,m,n} \models \mathsf{IP_{m,n,\epsilon}}] \geq 1 - \delta/2. \tag{37}$$

However, by virtue of Proposition B.1, for $n \geq \frac{8K_{u,\infty}^2 \theta^2 n_u \ln\left(\frac{8n_u}{\delta}\right)}{\epsilon^2}$ and $m \geq \frac{8K_{c,\infty}^2 \theta^2 n_c \ln\left(\frac{8n_c}{\delta}\right)}{(1-\gamma)^2 \epsilon^2}$

$$\langle \mu_{\nu_0}^{\pi_\mathrm{E}}, \tilde{c}_{N,m,n} - T_\gamma^* \tilde{u}_{N,m,n} \rangle$$
$$\leq \widehat{\langle \mu_{\nu_0}^{\pi_\mathrm{E}}, \tilde{c}_{N,m,n} \rangle} + \widehat{\langle \nu_0, \tilde{u}_{N,m,n} \rangle} + \epsilon \tag{38}$$

with probability $\mathbb{P}_{y,\tau,\xi}$ at least $1 - \delta/2$. Combining (37), (38) and the bounds in Theorem 2 of [76] by a simple union bound completes the proof. $\qquad\square$

# C Numerical Results

We consider a one-dimensional truncated Linear-Quadratic-Gaussian (LQG) control problem comprising a linear dynamical system

$$x_{t+1} = Ax_t + Ba_t + \omega_t, \quad t \in \mathbb{N},$$

and a quadratic state cost $c(s, a) = Qx^2 + Ra^2$, where $A, B \in \mathbb{R}, Q > 0, R > 0$. We assume that state and action spaces are given by $\mathcal{X} = \mathcal{A} = [-L, L]$ for some parameter $L > 0$. The disturbances $\{\omega_t\}_{t \in \mathbb{N}}$ are i.i.d. random variables generated by a truncated normal distribution with known parameters $\mu$ and $\sigma$, independent of the initial state $x_0$. Thus, the process $\omega_t$ has a distribution density

$$f(s, \mu, \sigma, L) = \begin{cases} \frac{\frac{1}{\sigma}\phi\left(\frac{s-\mu}{\sigma}\right)}{\Phi\left(\frac{L-\mu}{\sigma}\right) - \Phi\left(\frac{-L-\mu}{\sigma}\right)}, & s \in [-L, L] \\ 0 & \text{o.w. ,} \end{cases}$$

where $\phi$ is the probability density function of the standard normal distribution, and $\Phi$ is its cumulative distribution function. The transition kernel $\mathsf{P}$ has a density function $p(y \mid x, a)$, i.e., $\mathsf{P}(C \mid x, a) = \int_C p(y \mid x, a)\mathrm{d}y$ for all $C \in \mathcal{B}(\mathcal{X})$, that is given by

$$p(y \mid x, a) = f(y - Ax - Ba, \mu, \sigma, L).$$

In the special case that $L = +\infty$ the above problem represents the classical LQG problem, whose solution can be obtained via the algebraic Riccati equation [77, p. 372]. By referencing [31, Lemma 7.1], we can readily conclude that Assumption 2.1 holds in this context with specific constants

$$L_\mathsf{P} = \frac{2L \max\{A, B\}}{\sigma^2 \sqrt{2\pi} \left(\Phi\left(\frac{L-\mu}{\sigma}\right) - \Phi\left(\frac{-L-\mu}{\sigma}\right)\right)}, \quad L_c = \max\{Q, R\}2L.$$

As value function $u : \mathcal{X} \to \mathbb{R}$ we use a simple polynomial of degree 2 ($n_u = 3$) such that

$$u(x) = \sum_{i=1}^{n_u} \beta_i u_i(x), \quad u_i(x) = x^{i-1},$$

whereas the cost function $c : \mathcal{X} \times \mathcal{A} \to \mathbb{R}$ is approximated by the following weighted sum ($n_c = 9$)

$$c(x, a) = \sum_{i=1}^{n_c} \alpha_i c_i(x, a),$$

where $c_1(x, a) = 1$, $c_2(x, a) = x$, $c_3(x, a) = a$, $c_4(x, a) = xa$, $c_5(x, a) = x^2$, $c_6(x, a) = a^2$, $c_7(x, a) = x^2a$, $c_8(x, a) = xa^2$, $c_9(x, a) = x^2a^2$. For the simulation, the parameters are set as $L = 10$, $A = -1.5$, $B = 1$, $Q = R = 1$, $\mu = 0$, $\sigma = 1$, and $\gamma = 0.99$. The code for these experiments can be found at github.com/RAPACIRLCS/code. The experiments were run on a workstation with an AMD Ryzen 9 5950X CPU (16 cores) and 128GB of RAM.

**Sampled Inverse Program with known transition kernel.** In our first experiments highlighted in Figure 3, we focus on the sampled inverse program $\mathsf{SIP_N}$. More precisely, we solve the program $\mathsf{SIP_N}$ for various choices of sample sizes $N$ and denote its corresponding optimizers as $(\tilde{\alpha}_N, \tilde{\beta}_N, \tilde{\varepsilon}_N)$ and its extracted cost function as $\tilde{c}_N = \sum_{i=1}^{n_c} \tilde{\alpha}_{N_i} c_i$. Figure 3a shows the probability of the learnt cost function $\tilde{c}_N$ being $(\tilde{\varepsilon}_N + \epsilon)$-inverse feasible for various choices of $\epsilon$ and $N$. The plotted probability represents the empirical probability derived from 1000 experiments. It is evident that, for a constant parameter $\epsilon$, the likelihood of being inverse feasible grows with the increase of the sample size $N$. Additionally, for a constant sample size $N$, the probability of being inverse feasible increases as the parameter $\epsilon$ decreases. Both of these trends align with and support our theoretical findings as outlined in Theorem 4.1. Figure 3b shows the objective function of program $\mathsf{SIP_N}$ for various choices of sample sizes $N$. Since these are random programs, we plot the empirical average (solid line) and its corresponding standard deviations (shaded area) derived from 1000 independent experiments. As expected, the objective value $\tilde{\varepsilon}_N$ increases as a function of the sample size $N$. Figure 3c visualizes the theoretical sample complexity of Theorem 4.1, i.e., for various choices of $\delta$ and $\epsilon$, we plot the

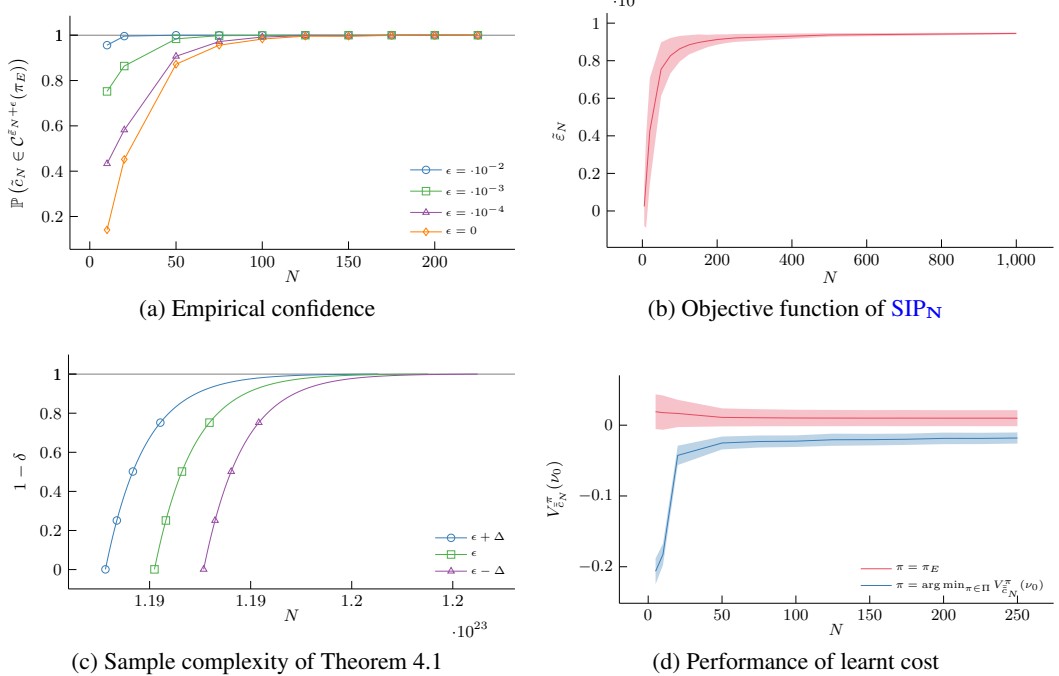

(a) Empirical confidence

(b) Objective function of SIP$_\mathbf{N}$

(c) Sample complexity of Theorem 4.1

(d) Performance of learnt cost

Figure 3: Solutions of the Sampled Inverse Program SIP$_\mathbf{N}$. The variable $N$ is the number of i.i.d. samples $(x, a)$ drawn uniformly from $\mathcal{X} \times \mathcal{A}$. We run 1000 independent experiments. Plot (a) shows the empirical probability of the estimated cost function $\tilde{c}_N$ being an element of the feasibility set, as described in Theorem 4.1 for given values of $N$ and $\epsilon$. Plot (b) shows the objective value of the random program SIP$_\mathbf{N}$, i.e., $\tilde{\varepsilon}_N$ on average over the 1000 experiments, where the shaded area shows the standard deviations. Plot (c) is a visualization of the theoretical sample complexity as given by Theorem 4.1. For various values of $\delta$ and $\epsilon$, we plot the sample size $N = \mathrm{N}(n_c + n_u + 1, g(\frac{\epsilon}{L_\Lambda}), \delta)$. The variation parameter is set to $\Delta = 1 \cdot 10^{-7}$. Plot (d) compares the discounted long-run costs $V_{\tilde{\tilde{c}}_N}^\pi(\nu_0)$ for the average $\bar{\tilde{c}}_N$ of the learnt costs $\tilde{c}_N$ under the expert policy $\pi_\mathrm{E}$ (red) and the optimal policy (blue). The solid line plots average over 1000 independent experiments, where the shaded area shows the standard deviations.

sample size $N = \mathrm{N}(n_c + n_u + 1, g(\frac{\epsilon}{L_\Lambda}), \delta)$. To simplify the computation, we used the closed form upper bound for the function N derived in [78] and given as

$$\mathrm{N}(n, \epsilon, \delta) \leq \frac{2}{\epsilon} \log\left(\frac{1}{\delta}\right) + 2n + \frac{2n}{\epsilon} \log\left(\frac{2}{\epsilon}\right).$$

When comparing Figure 3a and Figure 3c, we can see that there is a significant gap between the empirical and theoretical bounds. The dynamics of a variation in $\epsilon$, however, match the empirically observed behaviour. Figure 3d visualizes the performance of the learnt cost $\tilde{c}_N$ by comparing the discounted long-run cost of the expert policy under this learnt policy $V_{\tilde{c}_N}^{\pi_\mathrm{E}}(\nu_0)$ with its theoretical lower bound $\min_{\pi \in \Pi} V_{\tilde{c}_N}^\pi(\nu_0)$. According to Theorem 4.1 and Proposition 3.1 for large $N$ this difference vanishes with high probability, this behaviour can be observed in the plot. To reduce the computational effort replace in Figure 3d the learnt cost $\tilde{c}_N$ with its empirical average, denoted $\bar{\tilde{c}}_N$, taken over 1000 independent experiments. More precisely, for 1000 initial conditions $x_0 \sim \nu_0$ we plot $V_{\bar{\tilde{c}}_N}^{\pi_\mathrm{E}}(x_0)$ and the theoretical lower bound $\min_{\pi \in \Pi} V_{\bar{\tilde{c}}_N}^\pi(x_0)$.

**Sampled Inverse Program with unknown transition kernel.** The second experiment, Figure 4, solves the sampled inverse program SIP$_\mathbf{N,m,n,k}$ with unknown transition kernel. Compared to SIP$_\mathbf{N}$, since we assume the transition kernel to be unknown, the inequality constraints are based on sampled state transitions. The empirical probability, shown in Figure 4a, is derived from 1000 independent experiments. To decrease the degrees of freedom in the parameter selection we set $n = m = k$ for the simulations. The behaviour of the empirical confidence, observable in Figure 4a, follows the

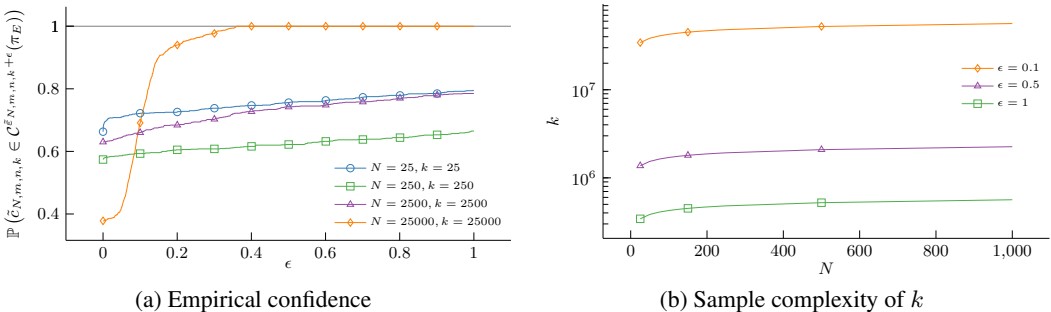

Figure 4: Solutions of the Sampled Inverse Program $\mathrm{SIP_{N,m,n,k}}$. The variable $N$ is the number of i.i.d. samples $(x, a)$ drawn uniformly from $\mathcal{X} \times \mathcal{A}$. We run 1000 independent experiments. Plot (a) shows the empirical probability of the estimated cost function $\tilde{c}_{N,m,n,k}$ being an element of the feasibility set, as described in Theorem 4.2 for different $N, k$ pairs given a chosen accuracy parameter $\epsilon$. Plot (b) shows the theoretical lower bound on $k$ depending on $N$, for a set $\epsilon$, as described by Theorem 4.2.

trends shown in Figure 3. As expected, an increase in the number of samples increases the confidence of learning a cost function $\tilde{c}_{N,m,n,k}$ that belongs to the inverse feasibility set $\mathcal{C}^{\tilde{\varepsilon}_{N,m,n,k}+\epsilon}(\pi_E)$. It can be seen how, even for the largest possible $\epsilon$, to reach a certain empirical confidence the $\mathrm{SIP_{N,m,n,k}}$ program requires many more state-action samples $N$ compared to the $\mathrm{SIP_N}$ program. When comparing the empirical confidence of a given $\epsilon$, $N$, and $k$ with the theoretical sample complexity, following Theorem 4.2, of $k$ corresponding to the same $\epsilon$ and $N$ it can be seen that the empirical sample performance of $\mathrm{SIP_{N,m,n,k}}$ is much more efficient.

