# OpenReview forum: "Randomized algorithms and PAC bounds for inverse reinforcement learning in continuous spaces"
_NeurIPS.cc/2024/Conference — NeurIPS 2024 poster_

### Official Review · Reviewer_PJ8y · 2024-07-10

**Soundness:** 3
**Presentation:** 1
**Contribution:** 2
**Rating:** 6
**Confidence:** 4

**Summary:**

This paper addresses inverse RL in discounted MDPs with continuous state and action spaces. First, the paper delves on the design of a suitable solution concept, which results in learning a reward that makes the expert at least approximately optimal plus a linear normalization constraint. Then, the paper studies the computational problem of extracting the latter solution with knowledge of the expert's policy and transitions. Finally, the paper provides a sample complexity result for the setting in which a dataset of trajectories collected by the expert and a simulator of the environment are available.

**Strengths:**

- The paper tackles a relevant problem that miss a theoretical understanding in the literature;
- The paper is base on an optimization perspective that is original with respect to prior works.

**Weaknesses:**

- The presentation is confusing and could be improved in many ways to make the paper more accessible;
- The paper provides mostly negative statistical results, despite the relatively strong assumptions. This is in contrast with the premise of the work to provide solutions for IRL in continuous state and action spaces;
- The motivation for some design choices (e.g., linear normalization constraint) and assumptions appears to be weak or absent.

**Questions:**

## Last comment after the author-reviewer discussion deadline

Dear Authors,

Sorry if my reply comes a little late on this. Having read the last replies, I wanted to get back to the manuscript and (briefly) to the recent papers (Metelli et al., 2021, 2023; Lindner et al., 2022; Lazzati et al., 2024a,b) on the reward feasible set formulation. I believe I have now a more robust understanding of the paper's contribution and I am changing my evaluation and confidence accordingly. A few words below to explain my reasons.

1)  Prior works have presented statistically efficient algorithms to learn the feasible reward set already. Especially, Metelli et al. (2023) did that for the tabular setting with generative model, Lazzati et al. (2024a) for tabular setting with offline data. The setting of this paper is slightly different (basically a mix of them, having offline demonstrations but online generative model). However, getting results for the latter setting from a combination of (Metelli et al., 2023; Lazzati et al., 2024) looks trivial. There is a very important caveat though: Their approaches are computationally intractable due to the nature of the feasible reward set. Lazzati et al. (2024) sidestep the issue by reframing the problem as a "reward checker". Given a reward, they have a tractable and efficient algorithm that can tell whether the reward belongs to the feasible set whp. This paper instead proposes an algorithm to provide a single reward belonging to the feasible set. Thus, the claim that this is the first work to solve the (reward learning) inverse problem in tabular MDPs with a tractable algorithm seems to be correct. This is a significant contribution in my opinion.
2) The main problem arise in the continuous state-action setting instead. The authors are saying that their expert's query and sample complexity are still polynomial for the considered "Lipschitz" MDP. However, to my understanding the number of sampled constraints should also be considered in the sample complexity, as they also require querying the generative model for one step. This makes the resulting sample complexity exponential in $dim (X \times A)$. The authors claim that the exponential dependence with the state and action dimensionality is inherent to the problem. Normally, in RL theory literature one would then prove a lower bound that support the claim to state that the problem is far-fetched, instead of claiming it is "solved" for low-dimensional settings.

Overall, I think the paper has valuable contributions and even the tabular results alone shall be communicated to the community. Other reasons for acceptance include a novel approach for reaching the tabular result and a nice concept of inverse feasibility. However, I'm changing my evaluation to only weak accept because most of the space in the paper is dedicated to a continuous space setting that appears to be mostly a dead end.

Some additional notes to the authors:
- Interestingly, the reward-feasibility notion seems to match the reward compatibility of the concurrent Lazzati et al. (2024b). The fact that two independent efforts reached a similar formulation may be further proof that this is the right direction;
- The discussion on the comparisons with previous papers above is extremely valuable and I suggest the authors to include that in a final version of the paper;
- While the authors may have a point that reward feasible set formulation is still ill-posed from a computational perspective, it looks well-posed to me in a statistical sense (which is arguably what matters the most);
- Having a deeper understanding of the paper required me quite some back-and-forth interactions with the authors. Perhaps some of the blame is on the presentation here, which could be made more accessible for a RL theory audience. I strongly encourage the authors to include the version of the bounds in the last comment in a final version.




----
----
----
----

This paper tackles an interesting problem in the recent wave of inverse RL with theoretical guarantees. While I have seen some recent works studying both IRL without online access to the expert's policy (Lazzati et al., Offline inverse RL, 2024) and IRL in continuous state spaces (Lazzati et al., How to scale RL to large state spaces? A provably efficient approach, 2024. Concurrent), this appears to be the first attempt at provably efficient IRL with continuous states and actions, as well as the first with this optimization perspective. Thus, I would tend to reward the paper for the novelty and originality. However, the provided results are mostly negative (even under strong assumptions, the number of samples required scales exponentially with the target accuracy) and they seems to leave IRL with continuous states and action as an open problem. This together with a cluttered presentation makes me lean towards a negative evaluation. There is a chance I am not understanding crucial parts of the submission, for which I encourage the authors to revise the presentation. I further provide some detailed comments below.

FUNCTION CLASS

The paper does not seem to commit to a specific (known) function class, such as linear MDPs, linear-mixture MDPs, etc. Trying to provide general results is definitely a feature, but not when generality hampers the ability to provide positive results (although in more restrictive settings).

INTERPRETABLE RESULTS

To my understanding, the paper provides essentially two main theoretical results: (1) approximation guarantees for computing a reward with knowledge of the expert's policy and transitions with a randomized program, (2) a sample complexity results with access to a dataset of trajectories obtained with the expert's policy and a generative model to estimate transitions. Both of the results are quite hard to interpret and a clear departure from how analogous results are presented in prior works. For instance, it would be important to understand the sample complexity as a function of the dimensionality of states and actions and other characteristics of the MDP. It is hard to extract those information from Th. 4.2.

SOLUTION CONCEPT

Instead of attempting at learning the feasible set of rewards, the paper proposes to add a linear normalization constraint. The latter choice is supported through Proposition 4.1, which assures that constant rewards can be avoided this way, but looks rather weak as a motivation in general. Aside from avoiding constant reward, we would like to be sure that the resulting reward is useful for a downstream application (such as one between the applications mentioned in the Introduction). The paper does very little to make a case in this sense.

On the same line, the Proposition 3.3 assures that the existence of at least one "good" reward in the feasible set. Do we have a similar guarantee for the reward resulting from the normalization constraint?

TYPOS

A few typos I noted while reading:
- l.73 "of"
- l.108 missing full stop
- l.254 "F" at start of the sentence

**Limitations:**

A section discussing limitations could have been included at the end of the paper.

---

> ### Author Rebuttal · Authors · 2024-08-07
>
> We thank the reviewer for the time spent in evaluating our work. In the following, we address your comments.
>
> ## Weaknesses
>
> > The presentation is confusing ...
>
> We would greatly appreciate it if you could specify which parts of the paper you find confusing. We put significant effort into presenting the material in an accessible and comprehensible manner without compromising on mathematical clarity and rigor. Given the constraints of the conference paper format and the substantial amount of theoretical concepts we cover, we aimed to present our findings concisely while maintaining coherence. For the final version of the paper, we will utilize the additional page allowed to further elaborate on certain sections and provide additional intuition and context about the presented theory.
>
> > The paper provides mostly negative statistical results ...
>
>  Could you please clarify what you mean? Additionally, we respectfully disagree with the comment that our assumptions are too strong. We believe that Ass. 2.1 in the setting of  continuous MDPs  is standard and, to the best of our knowledge, as general and mild as possible (see also lines 179-187). Ass. 4.1 is easily satisfied if X and A are compact and the sampling distribution continuous. Furthermore, Ass. 4.2 and 4.3 are directly satisfied if a tie-breaking rule is introduced, as noted in the text.
>
>
> >The motivation for some design choices ...
>
> The normalization constraint is motivated by an entire section (Sec. 4.1) and theoretically justified by Prop. 4.1. We are certainly open for suggestions on how to motivate it even better. Also it would be helpful if you could specify what "other design choices" you are referring to.
>
>
> ## Questions:
>
> > This paper tackles an interesting problem ...
>
> Thank you for the 2024 references, which we will add and discuss in our revised manuscript. We briefly highlight our main contributions and how they differ from recent theoretical works on IRL, underscoring the impact of our paper in this new research wave.
>
> First of all we provide a novel optimization perspective of the IRL problem that allows us to characterize the inverse feasibility set by means of Lagrangian duality. We believe this new formulation, applicable to both discrete and continuous MDPs, is more powerful and independent of the expert's complexity, unlike previous works [48-53] and Lazzati et al. 2024. Most importantly, it erases  the problems encountered in the formulations [51-53] and Lazzati et al. 2024.
> The authors in these works study solely how to deal with the unknown transition law and how to sample efficiently from the expert policy. In their formulation addressing these questions is complicated. However in our formulation this question is easily addressed with efficient expert sample complexity by just using Hoeffding bounds (Prop B.1 in the proof of Thm 4.2).
>
> Second, unlike recent theoretical IRL works [48-53] which avoid discussing this ill-posedness issue altogether, we attempt to address the ambiguity problem and provide theoretical results in this direction, hoping to lay the foundations for overcoming current limitations.
>
> Our result on linear function approximators (Prop. 4.2) is applicable even in the tabular setting considered in [48,49,51-53]. This contribution is crucial and missing from the literature on large-scale finite MDPs.
>
> Finally, we addres the problem of extracting one single cost function with theoretical guarantees. Learning a whole inifinite feasibility set is impossible even for the simplest tabular MDP problem. While constraint sampling has been proposed as a heuristic in the pioneering paper by Abbeel and Ng (2000), we provide explicit performance bounds and sample complexity for this methodology in continuous MDPs. The corresponding sample complexities include expert sample complexity, number of calls to the generator, and number of sampled constraints. The first two complexities scale gracefully with problem parameters, whereas the number of sampled constraints scales exponentially with the state and action space dimensions, making the algorithm particularly suitable for low-dimensional problems.
>
>
>
>
> > FUNCTION CLASS:
>
> Our paper tackles the inverse RL problem with minimal assumptions, contributing significantly to the limited theoretical research in this area. While some sample complexity bounds may be impractical, the results are not negative. The reviewer suggested that additional structures like linear MDPs could improve PAC bounds. Prior work supports this idea (see Zhang et al., 2015). We are interested in exploring this in future work, although it will require substantial effort.
> > INTERPRETABLE RESULTS:
>
> Thank you for your comment. Thm 4.1 ensures that the cost function $\tilde{c}_N$ from the finite-dimensional random convex problem $(SIP_N)$ is $(\tilde{\varepsilon}_N + \epsilon)$-inverse feasible for the expert policy $\pi_E$. Here, $\tilde{\varepsilon}_N$ is the solution to $(SIP_N)$, and $\epsilon$ is an accuracy parameter affecting the sample complexity $N$, which scales with $\{1/\epsilon, \log(1/\delta), n\}$ as described on Line 328. The sample complexity depends on MDP characteristics, such as state-action space dimension (exponentially) and basis functions for cost $n_c$ and value function $n_u$, as noted in Rem. 4.1. Thm 4.2 addresses a more realistic scenario without requiring this knowledge. We will clarify these results further in the additional page we are given.
>
>
> > SOLUTION CONCEPT:
>
> Our approach allows learning an approximately feasible reward in a PAC sense, as demonstrated by Theorems 4.1 and 4.2. While our normalization constraint prevents trivial solutions, our paper does not address selecting the "most informative" reward, which could be motivated by its utility for downstream tasks. Adding regularization constraints, such as an entropy term, could help select such a reward. This integration into our scenario programs is a topic for future research.

---

> > ### Comment · Reviewer_PJ8y · 2024-08-08
> >
> > Dear Authors,
> >
> > Thank you for considering the points raised in my review and for providing extensive replies. Let me first clarify that I made use of the "Weaknesses" box to summarize, in a few brief points, my evaluation of the paper. This is intended to help the AC in their decision and do not require rebuttal from the authors. Instead, I reported detailed comments in the "Questions" box covering points I would ask for clarifications.
> >
> > On the latter, I am following up below to better express some of my concerns, which I do not currently consider resolved. Then, I pointedly reply to some of the authors' comments.
> >
> > CONCERNS
> >
> > 1) Motivation
> >
> > The novel optimization perspective looks great, but I would like to understand what kind of benefits it provides on top of prior results. Does it allow for sharper rates in {expert's query complexity, sample complexity} or polynomial rates for previously unaddressed settings? Does it open the door for more scalable algorithms? Something else?
> >
> > 2) Interpretation
> >
> > Can the authors provide {expert's query complexity, sample complexity} rates in the form $O(S^x A^y (1 - \gamma)^{-k} \epsilon^{-z} ...)$ as it is common in tabular reinforcement learning literature? Can they provide analogous for the function approximation setting?
> >
> > 3) Solution concept
> >
> > Are the authors saying something on the lines of "since the inverse problem is ill-posed, we chose to learn a single reward selected arbitrarily (through normalization constraints) in the feasible reward set"? If so, then my concern is that this would make the problem well-posed, but it risks defeating the purpose of solving the problem itself: We want to learn a reward to make some use of it. I have to admit this problem is not fully solved by prior works either, but this is why some recent works make a conservative choice of learning all of the feasible rewards in the absence of any more reasonable criteria.
> >
> > REBUTTAL
> >
> > > The authors in these works study solely how to deal with the unknown transition law and how to sample efficiently from the expert policy.
> >
> > Isn't this what we care about in understanding the sample efficiency of inverse RL?
> >
> > > Second, unlike recent theoretical IRL works [48-53] which avoid discussing this ill-posedness issue altogether
> >
> > The papers [51-53] definitely discuss the ill-posedness
> >
> > > Our paper tackles the inverse RL problem with minimal assumptions
> >
> > Can the authors elaborate here? What does make their assumptions minimal? Ass. 2.1 appears to be rather strong: Does this include tabular MDPs? Would you provide references to support the claim that the assumption is standard?
> >
> > > The sample complexity depends on MDP characteristics, such as state-action space dimension (exponentially)
> >
> > Exponential sample complexity in state-action dimension is perceived as a negative result in reinforcement learning literature
> >
> > Best regards,
> >
> > Reply Author
> > Reviewer PJ8y

---

> > > ### Author Response · Authors · 2024-08-12
> > > **Clarifications on other raised points.**
> > >
> > > >The papers [51-53] definitely discuss the ill-posedness.
> > >
> > > We respectfully disagree with this point. The authors in [51-53] address the a*mbiguity problem* by estimating and learning the entire inverse feasibility set—essentially, the full set of cost functions that make the expert policy optimal. While this is an interesting theoretical challenge, it is impractical in reality. Even in the simplest finite MDP setting, with a known expert policy and transition law, computationally learning the entire inverse feasibility set is impossible. This set is uncountably infinite, and only a few classes of inverse-feasible costs can be represented analytically, such as trivial solutions of the form {${c\equiv C | C\in\mathbb{R}}$} and {$c=T_\gamma^*u | u\in\mathbb{R}^{|X|}$}.
> > >
> > > Please note that in"How to Scale IRL to Large State Spaces? A Provably Efficient Approach" (2024 Concurrent), the authors explicitly say (page 5):
> > >
> > > *Limitations of the Feasible Set. Lack of Practical Implementability: It contains
> > > a continuum of rewards, thus, no practical algorithm can explicitly compute it.*
> > >
> > > >The authors in these works study solely how to deal with the unknown transition law and how to sample efficiently from the expert policy. Isn't this what we care about in understanding the sample efficiency of inverse RL?
> > >
> > > If one aims to estimate only the entire inverse feasibility set, there are two primary sources of error: estimation error due to the unknown expert policy and estimation error due to the unknown transition law. Consequently, the relevant sample complexities are the expert sample complexity and either the number of queries to the generative oracle (for the generative model) or the number of episodes (for the forward model).
> > >
> > > However, if the goal is to extract a single cost function that is $\varepsilon$-inverse feasible for large-scale finite MDPs, we must manage  a feasibility problem of $|X||A|$ constraints and variables of dimension $|X||A|$. This challenge also applies to the formulations in [50-53]. For continuous MDPs, the problem becomes even more complex due to its infinite-dimensional nature. In addition to the earlier sources of error, we must also account for approximation error due to function approximation and optimization error in solving the feasibility problem. This results in additional iteration complexity if using gradient-based algorithms or an the number of sampled constraints if choosing constraint sampling techniques.
> > >
> > > This theoretical challenge is not addressed in [51-53] or in the new 2024 submissions. We emphasize that for small finite tabular MDPs, one can simply solve a finite linear program in polynomial time.
> > >
> > > >The sample complexity depends on MDP characteristics, such as state-action space dimension (exponentially)
> > > Exponential sample complexity in state-action dimension is perceived as a negative result in reinforcement learning literature
> > >
> > > First of all, we would like to highlight that the corresponging bound is a part of the the big amount of theoretical concepts covered by the paper.
> > >
> > > The exponential growth in sample complexity with respect  to the dimensions of the state and action space, as demonstrated in our paper, is an unavoidable consequence of Bellman's curse of dimensionality. Given the generality of our bounds, this growth is inherent. However, it's important to note that even with this exponential growth, settings with low-dimensional MDPs can still achieve practically useful convergence, as evidenced by our  theory and numerical experiments on the one-dimensional LQR problem.
> > >
> > > In contrast to heuristic IRL algorithms for continuous state and action spaces, where no guarantees are provided, our work proves convergence and offers explicit bounds. While our bounds for high-dimensional MDPs may not yet be practically useful, we view this paper as a crucial first step toward developing provably efficient algorithms for continuous IRL problems.
> > >
> > > Furthermore, as we’ve already discussed, the works [51-53] and the recent 2024 contributions do not address the extraction of a cost function, i.e., solving a large-scale, even infinite-dimensional feasibility problem for continuous MDPs. In these cases, the exponential growth in the number of constraints does not appear in the finite sample analysis.
> > >
> > > Finally, the only work that extracts a single cost function in continuous state and discrete action spaces, [50], also suffers from exponential dependence on the dimensionality of the state space.

---

> > > ### Author Response · Authors · 2024-08-12
> > > **Clarification on other raised points (Cont'd)**
> > >
> > > > Can the authors elaborate here? What does make their assumptions minimal? Ass. 2.1 appears to be rather strong: Does this include tabular MDPs? Would you provide references to support the claim that the assumption is standard?
> > >
> > > For finite discounted MDPs the existence of an optimal policy and the Bellman optimality equation hold automatically. However, in continous MDPs this is not the case.
> > >
> > > Assumption 2.1 involves the usual continuity-compactness conditions [ 56 ]  -which ensure the existence of an optimal policy and the Bellman optimality equation - together with the Lipschitz continuity of the
> > > elements of the MDP (see, e.g., [57]) - which ensures the Lipschitz continuity of optimal value function.  The Lipschitz continuity assumption encompasses various probability
> > > distributions, such as the uniform, Gaussian, exponential, Beta, Gamma, and Laplace distributions, among others.
> > > Consequently, Assumption 2.1 accommodates a broad range of MDP models and allows for the
> > > consideration of smooth and continuous dynamics that reflect the characteristics of several real-world
> > > applications, such as robotics, or autonomous driving. (see lines 179-187)
> > >
> > > In the tabular MDP setting, our analysis holds without any additional assumptions on the MDP model. This is because we bypass the need for sampling constraints by solving a finite linear program using off-the-shelf solvers. In contrast, [48, 49] assume that the problem is $\beta$-separable and that every state is reachable with probability $\alpha$. For the MDP with a continuous state space in [50], the authors assume that the transition law density has an infinite matrix representation. Our Assumption 2.1 on the MDP is significantly milder and includes the representation considered in [50] as a special case.
> > >
> > > > Interpretation: Can the authors provide {expert's query complexity, sample complexity} rates in the form $O(S^x A^y (1 - \gamma)^{-k} \epsilon^{-z} ...)& as it is common in tabular reinforcement learning literature? Can they provide analogous for the function approximation setting?
> > >
> > >
> > > For tabular MDPs, offline access to expert for tabular MDPs and a generative model we require  $m = \mathcal{O}\left(\frac{|X||A|(\log(\frac{|X||A|}{\delta})}{(1-\gamma)^2\varepsilon^2}\right)$ expert samples and $K|X||A| = \mathcal{O}\left(\frac{X|^2|A|\log(\frac{|X|^2|A|}{\delta})}{\varepsilon^2}\right)$ calls to the generative model and solve the resulted sampled finite LP with $|XA|+|X|$ variables and $|X|$ constraints in polynomial time, to learn a cost function that is $\varepsilon$-inverse feasible with probability $1-\delta$.
> > >
> > >
> > > For continous MDPs when we use $n_u$ basis functions for the value function and $n_c$ basis functions for the cost function we need $m = \mathcal{O}\left(\frac{n_c\log(\frac{n_c}{\delta})}{(1-\gamma)^2\varepsilon^2}\right)$ expert samples and $K = \mathcal{O}\left(\frac{n_u\log(\frac{n_u N}{\delta})}{\varepsilon^2}\right)$ calls to the generative model per constraint and $N=\mathcal{O}(\exp^{dim(X\times A)})$ sampled constraints and and solve the resulted sampled finite LP with $n_u+n_c$ variables and $N$ constraints in polynomial time to learn a cost that is $\varepsilon+\varepsilon_{\textup{approx}}$-inverse feasible with probability $1-\delta$
> > >
> > > Note that as we argued in detail above our setting is different from the one in [51-53] since we have offline access to the expert and address a different question, i.e. learning a single reward with formal guarantees in continuous MDPs. For the PAC learning scenario considered in [51-53] our bounds translate in $m = \mathcal{O}\left(\frac{|X||A|(2-\gamma)^2\log(\frac{|X||A|}{\delta})}{(1-\gamma)^4\varepsilon^2}\right)$ expert samples and $K|X||A| = \mathcal{O}\left(\frac{(2-\gamma)^2|X|^2|A|\log(\frac{|X|^2|A|}{\delta})}{(1-\gamma)^2\varepsilon^2}\right)$ calls to the generative model. There is no need to solve the  sampled program. The factors $\frac{2-\gamma}{1-\gamma}$ arise from Prop. 3.3.
> > >
> > > > Solution concept
> > >
> > > Overall, the purpose of the normalization constraint is to exclude a large class of meaningless solutions. In particular, we state and prove that the normalization constraint rules out a wide class of trivial solutions, i.e., all constant functions and inverse solutions of the form  $c=T_\gamma^* u$, an outcome devoid of physical meaning and a mathematical artifact. The recovered cost is $\varepsilon+\varepsilon_{\textup{approx}}$-inverse feasible with probability $1-\delta$. The interpretation of this guarantee is given in Prop. 3.1 and 3.3. In particular, note that by adopting the terminology of Lazzati et al. 2024 the recovered cost is  $\varepsilon+\varepsilon_{\textup{approx}}$- compatible with respect to the expert. Investigating how to define different normalization constraints and the implications for the resulting learned cost is a topic of future research.

---

> > > ### Author Response · Authors · 2024-08-14
> > >
> > > Dear Reviewer,
> > >
> > > As the discussion window is closing soon, we wanted to check if you have any remaining questions or concerns. We would be happy to provide further clarification on any points that may need additional discussion.
> > >
> > > Thank you again for your time and valuable feedback.
> > >
> > > Best regards,
> > > The Authors

---

> ### Author Response · Authors · 2024-08-12
> **Detailed comparison to prior results**
>
> Thank you for your prompt response and for providing us with the opportunity to further clarify our points. Below, we address the issues you raised.
>
> >CONCERNS:
> > Motivation: The novel optimization perspective looks great, but I would like to understand what kind of benefits it provides on top of prior results. Does it allow for sharper rates in {expert's query complexity, sample complexity} or polynomial rates for previously unaddressed settings? Does it open the door for more scalable algorithms? Something else?
>
> Recent research in IRL with theoretical guarantees has seen significant advancements, as evidenced by works such as [48,49,50,51,52,53], Lazzati et al., "Offline Inverse RL" (ICML 2024), and Lazzati et al., "How to Scale RL to Large State Spaces? A Provably Efficient Approach" (2024). These studies have laid the groundwork for exploring the mathematical foundations of IRL, a complex and challenging topic. In the following, we will clarify the specific benefits our optimization-based perspective and formulation provide on top of prior results.
>
> First, as discussed in our manuscript and demonstrated in Appendix A.2, our formulation—when applied to finite tabular Markov decision processes (MDPs) and a stationary Markov expert policy—simplifies to those considered in [48-53].
>
>
> **Comparison to [48,49,50]**
>
> *Setting*: The formulation underlying the analysis and algorithms in [48, 49, 50] addresses scenarios where the expert policy is known and deterministic. In contrast, our formulation accommodates more general unknown, nonstationary, and randomized expert policies. Additionally, our approach assumes access to an offline, static dataset of expert demonstrations. While the works in [48, 49] focus on tabular MDPs, and [50] addresses MDPs with continuous state and discrete action spaces, we extend these contributions by providing a formulation and formal guarantees for IRL in continuous state and action spaces.
>
> *Formal Guarantees*: Notably, similar to our Theorems 4.1 and 4.2, the sample complexity in the continuous-state setting, as provided in [50], scales exponentially with the dimension of the state space.
>
> *Assumptions*: In the tabular MDP setting, our analysis holds without any additional assumptions on the MDP model. This is because we bypass the need for sampling constraints by solving a finite linear program using off-the-shelf solvers. In contrast, [48, 49] assume that the problem is $\beta$-separable and that every state is reachable with probability $\alpha$. For the MDP with a continuous state space in [50], the authors assume that the transition law density has an infinite matrix representation. Our Assumption 2.1 on the MDP is significantly milder, as it encompasses the majority of probability distributions (see lines 179-187) and includes the representation considered in [50] as a special case.

---

> ### Author Response · Authors · 2024-08-12
> **Detailed comparison to prior works (Cont'd)**
>
> **Comparison to [51,52,53]**
>
> The characterization of the inverse feasibility set which forms the basis for the analysis in [51,52,53] is the following:
>
> Let $\pi_E$ be stationary Markov. A cost function $c$ is inverse if and only if there exists $u\in\mathbb{R}^{|X||A|}$ and nonnegative $L\in{\mathbb{R}}^{|X||A|}$ such that $c(x,a)-u(x)+\gamma\mathbb{E}_{y\sim P(\cdot|x,a)}[u(y)] = L(x,a)\mathbb{1}[\pi_E(a|x)=0]$, for all $(x,a)\in X\times A$.
>
> This formulation considers a stationary Markov expert, whereas our proposed approach is independent of the complexity of $\pi_{E}$. Their formulation cannot be extended to MDPs with continuous state and action spaces due to the presence of the term $\mathbb{1}[\pi_E(a|x)=0]$. For continuous distributions, $\pi_E(a|x)=0$ always, making the extension infeasible. Additionally, the term $\mathbb{1}[\pi_E(a|x)=0]$ significantly complicates the estimation of the inverse feasibility set. To address this challenge, they assume active querying of $\pi_E$. In contrast, our optimization-based framework formulates the problem using occupancy measures instead of policies. This allows us to tackle the more challenging learning scenario where the learner has access only to a finite set of expert trajectories and is not permitted to query the expert for additional data during training. In our formulation, efficiently estimating the term $\widehat{<\mu^{\pi_{\textup{E}}}_{\nu_0},c>}$ via sample averages is straightforward (see the sampling process in lines 361-367), and the estimation error can be computed using a simple Hoeffding bound.
>
> In [51, 52, 53], the authors focus exclusively on estimating the inverse feasibility set, bypassing the task of learning a single cost with formal guarantees. For the tabular setting with a generative model and an (online) interactive expert, [53] provides a sample complexity lower bound of $\mathcal{O}\left(\frac{|X||A| (|X|+\log(\frac{1}{\delta}))}{(1-\gamma)^3 \varepsilon^2}\right)$ and proposes a sampling strategy that achieves this bound up to logarithmic factors. It is important to note that when estimating the inverse feasibility set, the sample complexities of interest are the number of queries to the generative model and the expert sample complexity. In other words, in our Thm 4.2, the number of sampled constraints required to solve the reduced finite LP for extracting a single cost is not necessary (recall that the curse of dimensionality appears only in the number of sampled constraints).
>
> In our offline setting for tabular MDPs, we require $m = \mathcal{O}\left(\frac{|X||A|(2-\gamma)^2\log(\frac{|X||A|}{\delta})}{(1-\gamma)^4\varepsilon^2}\right)$ expert samples and $K|X||A| = \mathcal{O}\left(\frac{(2-\gamma)^2|X|^2|A|\log(\frac{|X|^2|A|}{\delta})}{(1-\gamma)^2\varepsilon^2}\right)$ calls to the generative model. The factors $\frac{2-\gamma}{1-\gamma}$ arise from Prop. 3.1 and 3.3.
>
> Furthermore, we address the challenge of extracting a single cost with formal guarantees for continuous MDPs by incorporating a normalization constraint, employing function approximation, and using constraint sampling techniques. For tabular MDPs, this task is straightforward, as one only needs to solve a finite MDP using off-the-shelf solvers in polynomial time.
>
> **Comparison to "How to Scale IRL to Large State Spaces? A Provably Efficient Approach" (2024 Concurrent)**
>
> The authors consider the setting of linear MDPs, where the agent can query the expert in any state and interact online with the environment (forward model). By generalizing the notion of the inverse feasibility set and introducing the *cost compatibility* framework, they provide a provably efficient algorithm for MDPs with continuous states and discrete actions to estimate the compatibility of all costs with high accuracy.
>
> Interestingly, in our versatile formulation, any cost that is $\varepsilon$-inverse feasible is also $\frac{1-\gamma}{2-\gamma}\varepsilon$-compatible (Prop. 3.1). This key feature, which seems absent and unaddressed in the formulations of [51-53], highlights another significant distinction in our approach. We conjecture that by employing the same Ridge regression estimators as in [Jin et al., "Provably Efficient RL with Linear Function Approximation" (2020)], one could establish similar efficient bounds for continuous $X$ and $A$. We leave this exploration for future work. We also emphasize that, akin to estimating the inverse feasibility set, this new theoretical question does not require sampling constraints.
>
> **Comparison to "Offline IRL" (ICML 2024), and Lazzati et al.**
>
> This setting contrasts with our assumption of a generative model.
>
> **A new perspective**
>
> Our optimization-based approach results in problem formulations directly amenable to modern large-scale stochastic optimization methods. Consequently, we anticipate that our techniques will benefit future algorithm designers and establish a foundation for more comprehensive research in this area.

---

### Official Review · Reviewer_579s · 2024-07-12

**Soundness:** 2
**Presentation:** 3
**Contribution:** 3
**Rating:** 6
**Confidence:** 2

**Summary:**

The paper proposes an optimization formulation of the inverse RL problem. The paper discusses the properties of the solution to the optimization problem. It further discusses how to reduce of the raw problem to a computation feasible one. It gives the theoretical analysis on the approximation error and sample complexity.

**Strengths:**

The paper proposes a detailed and rigorous description of the problem setting, the method and the results. The contributions are clearly stated and the overall writing is good and easy to follow. The paper provides both theoretical results and experiments, which makes the discussion relatively complete.

**Weaknesses:**

1. The paper's result lacks benchmark. First, it lacks comparison with existing results, which makes it hard to evaluate the improvement made by the paper. Second, the paper doesn't reduce its general result to simpler special cases, like tabular or linear MDPs, so it's hard to get an idea how the result looks like on these more familiar examples. Third, the paper doesn't give insights on the hardness of the problem, namely, a lower bound. Without these benchmarks, the first impression is the bounds given by the paper are very loose, which heavily depends on $\Vert\cdot\Vert_\infty$ or $\Vert\cdot\Vert_L$ norms.

2. Part of the result is unintuitive. In Proposition 4.2, the notation $\min\{1, d\}$ seems to imply the possibility $d < 1$, which seems to be unintuitive. Can the authors give a more detailed explanation on it? And the bound shows that the bigger $\theta$ is, the better the result is. In this case, what's the point of introducing $\theta$, instead of directly letting it be $\infty$?

**Questions:**

1. For challenge (a), it seems the paper's solution only excludes constant functions. There should still be ambiguity in the optimal solution. In this case, how do the authors ensure $c^\star, u^\star$ in the later sections are well-defined?

2. For figure (c) in the experiment, the $x$-axis looks to be extremely large, with the values almost not changed on the whole axis. And the value $\delta$ also becomes extremely small. What's the value of $\varepsilon$ in this figure and what happens if one sets $\varepsilon$ values to be the same as in figure (a)?

**Limitations:**

/

---

> ### Author Rebuttal · Authors · 2024-08-07
>
> We thank the reviewer for the time spent in evaluating our work. In the following, we address your comments.
>
> ## Weaknesses
>
> > The paper's result lacks benchmark ...
>
> This work focuses on theoretical aspects of inverse RL, specifically on learning a cost function in continuous state and action spaces from observed optimal behavior, and deriving PAC bounds. We do this by approximating the infinite-dimensional problem with a finite convex problem and rigorously quantifying approximation errors.
>
> Most existing inverse RL research is on finite state and action spaces. Current work on continuous spaces either lacks theoretical guarantees or focuses on the feasibility set (see [42, 43, 44, 23, 26, 45, 27], [51, 52, 53]). Our related work section offers a detailed comparison. We also refer the reviewer to our response to Reviewer PJ8y (in section "Questions"), where we briefly highlight our main contributions and how they differ from recent theoretical works on IRL, underscoring the impact of our paper in this new research wave.
>
> While we demonstrate our method with a simple truncated LQR example (see Appendix C), we agree that comparing it with existing heuristic methods in continuous spaces is valuable. Our one-dimensional experiment underscores the need for further investigation into choosing basis functions and sampling informative state-action pairs. We are addressing these questions theoretically and will include a discussion in the revised paper.
>
> >Second, the paper doesn't reduce its general result to simpler special cases ...
>
> Thank you for the comment. In the tabular MDP case, our proposed bounds become redundant. If the transition kernel $T$ is known, the $\varepsilon$-inverse feasibility set (Equation 1) is a linear program with finite variables, solvable in polynomial time. Thus, our method for approximating infinite-dimensional LPs with function approximation and constraint sampling is unnecessary, making our error bounds for this approximation irrelevant.
>
> Our approach is intended for continuous MDPs. For large but finite state and action spaces, where direct solution of LPs is impractical, function approximation and constraint sampling are applicable, and our PAC bounds are relevant.
>
> Regarding additional structure, such as a linear MDP, related works suggest that incorporating such structures could improve PAC bounds (see [2]). Investigating how to leverage linear MDP structures in our inverse RL approximation hierarchy is a promising future direction, though it requires considerable effort.
>
> [2] Zhang et. al., On the sample size of random convex programs with structured dependence on the uncertainty, Automatica, Volume 60, 2015
>
> >Part of the result is unintuitive ...
>
> We thank the reviewer for this comment. As written, it is indeed unintuitive because there is a small typo causing the confusion. The constant $d$ in Proposition 4.2 should be $d = leb(\mathcal{X} \times \mathcal{A})$, where $leb(\cdot)$ denotes the Lebesgue measure. In the proof, on line 671, it is clear why $leb(\mathcal{X} \times \mathcal{A})$ is the correct constant.
>
> >And the bound shows that ...
>
> Indeed, the reviewer is correct. When examining the bound in Proposition 4.2, the resulting error monotonically decreases as $\theta$ increases, suggesting that $\theta = \infty$ would be optimal. However, the sample complexity in Theorem 4.1 increases monotonically with $\theta$. In other words, a larger $\theta$ requires a larger number of constraint samples. Therefore, $\theta$ must be selected to balance these two factors. We will include a remark on this in the revised version.
>
> ## Questions:
>
> > 1.For challenge (a), it seems ...
>
> This is an assumption. We assume that the expert policy $\pi_E$ is optimal for a unique true cost function $c^\star$ with $u^\star$ the corresponding optimal value function.
> Although this true cost is unknown, some prior knowledge about its properties allows us to choose appropriate linear function approximators to make the projection residuals in the theorem sufficiently small. For example, if the true cost function is known to be smooth, Fourier or polynomial basis functions can be used. Essentially, the term $\varepsilon_{\textup{approx}}$ measures the expressiveness of the linear function approximators.
>
> We would like to highlight that, in practice, assuming the expert policy is produced by a unique true cost does not contradict the ill-posed nature of the IRL problem, where the expert policy is optimal for infinitely many cost functions.
>
> A key question is whether, despite assuming a unique true cost, any inverse feasible cost solution can replace the true cost in Proposition 4.2's approximation error expression due to IRL's inherent ambiguity. The answer is yes: if the basis functions approximate the true cost well, it will be recovered accurately. Otherwise, if the approximators represent a different inverse feasible cost, that cost will be recovered instead. This aligns with expectations.
>
> We’ve added this discussion to the revised paper for clarity. Thank you for highlighting this point.
>
> > For figure (c) in the experiment ...
>
> Thank you for your comment. Figure (c) shows how sample complexity (x-axis) depends on the confidence level $1-\delta$ (y-axis). As sample size increases, the confidence parameter $\delta$ decreases, which aligns with the logarithmic growth of sample complexity with $\log(1/\delta)$.
>
> Figure (a) evaluates empirical confidence with four different accuracy parameters. As noted, the parameters result in impractically large sample complexity bounds. The proposed bounds have large constants, limiting their practical use compared to the empirical results in figures (a), (b), and (d). The main value of these bounds is in understanding how sample complexity grows with problem parameters like accuracy $\epsilon$ confidence $\delta$ and dimensions.
>
> Additional examples of figure (c) with different parameters are provided in the supplementary PDF.

---

### Official Review · Reviewer_sQPr · 2024-07-13

**Soundness:** 2
**Presentation:** 2
**Contribution:** 2
**Rating:** 4
**Confidence:** 2

**Summary:**

This paper investigates the problem of inferring cost functions from observed optimal behavior in continuous state Markov Decision Processes. The authors develop their theoretical framework initially assuming full access to the expert policy. To address the issue of trivial solutions, they introduce a linear normalization constraint. The study then progresses to a more practical scenario, providing error bounds for cost estimation when working with limited expert samples and a generative model. This approach bridges the gap between theoretical analysis and practical applications in inverse reinforcement learning for continuous domains.

**Strengths:**

- Avoiding repeated RL solving: Unlike many existing methods, this approach does not rely on repeatedly solving the forward reinforcement learning problem, which is computationally expensive for continuous spaces MDP.

- Sounding Theoretical guarantees: The paper provides probabilistic convergence guarantees on the quality of the recovered solution, bridging the gap between theory and practice in continuous IRL.

- Addressing reward ambiguity: The paper contributes to tackling the reward ambiguity problem by adding a normalization term.

**Weaknesses:**

- The paper appears to present a series of theorems without providing sufficient intuitive explanations or justifications. This approach can make the work difficult to understand for reviewer.

- Despite the theoretical rigor, the paper seems to lack substantial practical demonstrations or empirical results. This absence makes it challenging to assess the real-world applicability.

- The algorithms proposed might be theoretically sound but potentially impractical for real-world applications. The paper may not adequately address computational complexity or scalability issues that could arise when applying these methods to large-scale or complex problems.

**Questions:**

See Weakness

---

> ### Author Rebuttal · Authors · 2024-08-07
>
> > The paper appears to present a series of theorems without providing sufficient intuitive explanations or justifications. This approach can make the work difficult to understand for reviewer.
>
>
>
> We put significant effort into presenting the material in an accessible and comprehensible manner without compromising on mathematical clarity and rigor. Given the constraints of the conference paper format and the substantial amount of theoretical concepts we cover, we aimed to present our findings concisely while maintaining coherence. For the final version of the paper, we will utilize the additional page allowed to further elaborate on certain sections and provide additional intuition and context about the presented theory. In particular, we will make an effort to explain and justify the two theorems of our paper better.
>
> Our first theorem (Theorem 4.1) guarantees that the cost function $\tilde{c}_N$, obtained from the solution of the proposed finite-dimensional random convex counterpart ($SIP_N$), is indeed $(\tilde{\varepsilon}_N + \epsilon)$-inverse feasible for the expert policy $\pi_E$. Here, $\tilde{\varepsilon}_N$ is the solution to $(SIP_N)$ and $\epsilon$ is an accuracy parameter
> affecting the sample complexity $N$, which scales with $\{1/\epsilon, \log(1/\delta), n\}$ as described on Line 328. The sample complexity depends on MDP characteristics, such as state-action space dimension (exponentially) and basis functions for cost $n_c$ and value function $n_u$, as noted in Rem. 4.1.
>
> Theorem 4.1 is primarily of practical interest because the program $(SIP_N)$ requires knowledge of the transition kernel, which is typically not available in RL. Therefore, our second theorem (Theorem 4.2) considers a more realistic setting where knowledge of the transition kernel is not required. In this setting, the proposed cost $\tilde{c}_w$ for $w=\{N,m,n,k\}$ is derived from the program $(SIP_w)$ for $w=\{N,m,n,k\}$. Essentially, Theorem 4.2 extends the results of Theorem 4.1 to this more realistic scenario.
>
>
> > Despite the theoretical rigor, the paper seems to lack substantial practical demonstrations or empirical results. This absence makes it challenging to assess the real-world applicability.
>
> We illustrate our method with a simple truncated Linear Quadratic Regulator (LQR) example (see App. C) to provide better intuition about the method and the proposed sample complexity bounds. The focus of our paper is on an optimization perspective of IRL and deriving fundamental, provable PAC bounds for continuous inverse RL in a very general setting. Thus, studying the empirical performance beyond the academic LQR problem is out of scope and will be addressed in future work. It is worth noting that the work of Dexter et al. [50], the only theoretical study addressing IRL in continuous state (but discrete action) spaces, also includes simulations for a representative one-dimensional state space MDP.
>
>
>
> > The algorithms proposed might be theoretically sound but potentially impractical for real-world applications. The paper may not adequately address computational complexity or scalability issues that could arise when applying these methods to large-scale or complex problems.
>
>
> The theoretical soundness of the proposed algorithm is ensured by our main results (Theorems 4.1 and 4.2). If "large-scale" is interpreted as the number of state/action variables, we address settings with continuous (uncountable) state and action spaces. Thus, even our simple 1-dimensional LQR example might be considered "large-scale." However, if "large-scale" refers to the dimension of the continuous state and action spaces, our bounds indeed suffer from exponential sample complexity in these dimensions. This is a manifestation of the infamous curse of dimensionality, which cannot be avoided. Please refer to our detailed discussion in Remark 4.1 of the paper.
>
> We agree with the reviewer that to apply our bounds to complex and high-dimensional real-world problems, we need to more carefully exploit underlying structures. Otherwise, the sample complexity bounds are too large and of limited practical use. However, we would like to emphasize that these sample complexity bounds are often conservative in practice, as can be seen in our empirical evaluation in the numerical results, see Section C. Experiments conducted on closely related scenarios have shown promising approximation bounds, as detailed in [1,2].
>
> In this work, our aim is to address the most general setting possible with the minimal assumptions required to establish PAC bounds for inverse RL. We plan to explore exploiting additional structures to derive tighter bounds for well-structured subclasses of problems in future work.
>
> Finally, we would like to emphasize that the provided bounds are part of our contributions. We refer the reviewer to our response to Reviewer PJ8y (in section "Questions"), where we briefly highlight our main contributions and how they differ from recent theoretical works on IRL, underscoring the impact of our paper in this new research wave.
>
>
> [1] Marco Campi and Simone Garatti, Introduction to the Scenario Approach, Society for Industrial and Applied Mathematics, 2018
> [2] Xiaojing Zhang, Sergio Grammatico, Georg Schildbach, Paul Goulart, John Lygeros, On the sample size of random convex programs with structured dependence on the uncertainty, Automatica, Volume 60, 2015

---

### Official Review · Reviewer_p8oS · 2024-07-14

**Soundness:** 4
**Presentation:** 4
**Contribution:** 4
**Rating:** 7
**Confidence:** 4

**Summary:**

The paper
1) establishes a formal framework for studying the problem of continuous state- and action-space inverse reinforcement learning (i.e. given an expert policy, recovering a set of cost functions for which the policy is optimal)
2) provides theoretical results characterizing the set of cost functions that would be recovered based on perfect knowledge of the expert policy assuming the ability solve an infinite-dimensional linear feasibility problem
3) provides a method for approximating the the infinite-dimensional problem using scenario approach to reduce the dimensionality of the problem, making the theoretical model computationally feasible
4) provides PAC results for the case where we only have access to a finite set of samples from the expert policy

**Strengths:**

The optimal control and reinforcement learning communities (i.e. also communities studying the "inverse optimal control" and "inverse reinforcement learning" (IRL) problems) have been living somewhat separate lives, each using their own formalisms and solution methods. I think there is a lot the two can learn from each other so I'm glad to see this paper, drawing heavily on the control-theory angle, at NeurIPS that has been more a harbour for the RL community.

I think the paper is great step toward putting work on IRL on firmer theoretical foundations. Almost all previous comparable cases of theoretical analysis in IRL that I've seen are addressing only the tabular case of finite state and action spaces which are of limited practical value. Extending this kind of theoretical treatment to continuous spaces is valuable. The paper does a great first step in that (and a sizeable one).

The paper tries to do a lot, but especially given the amount of theoretical concepts it covers, it manages to maintain good clarity, with a minimal amount of typos and other mistakes. (I want to emphasize my appreciation for the level of writing here - I can easily imagine a paper covering the same technical ground which would be hell to read for someone previously unfamiliar with some of the theory. It must have been a lot of work to rewrite it this way: I do appreciate it and think it makes for a much more impactful paper!)

**Weaknesses:**

- The paper certainly contains enough material for a substantially longer journal paper - some of the sections could benefit from being longer, trying to convey more intuitions and context about the presented theory. More thorough empirical evaluation ( /illustration) would also be beneficial to give readers a better intuitions for what are the actual numbers involved and what's the computational scalability beyond the simple example provided. That said, this is infeasible within the scope of a single conference paper on top of all that the authors already did and on balance, I personally think this paper is still a worthy contribution to NeurIPS readership as is.
- The bounds provided by the theorems don't always seem meaningful. In the only example presented suggests >10^23 samples are needed to get guarantees - and that is on an super-simple problem with a 1D state-space and 1-D action space! But other theorems seem to provide more meaningful results, and I do consider the work presented a good starting point for further analysis.

**Questions:**

- In Section 4.1 you claim that "the inverse feasibility set C(πE) contains some solutions that arise
from mathematical artifact". Sure, translation by a constant can be considered an artifact, but is this the case for all potential-shaping? Isn't the identifiability issue a deeper issue than a mere "mathematical artifact"? On a fixed environment, optimal policies for meaningfully different rewards may happen to coincide due to the environment structure; however, change the environment a bit (e.g. by changing the transition kernel), and the optimal policies may no longer coincide (see e.g. work on "transferability" or rewards learnt via IRL). It's a frequently raised selling point of IRL that the reward is a more generalizable representation of the goal than a policy.
- Do you have an intuitive explanation for why the constraint on the epsilon constraint on the consistency of the cost and value function is enforced point-wise from below and in expectation from above?

Minor points and suggestions:
- In the first paragraph, I don't fully understand your third motivation for inverse decision making: I suggest either explaining it better (and how it meaningfully differs from motivation 2) or removing it to prevent confusion.
- There seems to be minor tension between the end of the initial introduction, ending by emphasizing "especially in safety-critical systems with potential fatal consequences", and then just after that, in the second sentence of contributions, you "emphasize that we are interested in situation where the cost function is of interest in itself". The first point seem to point in the direction of using the recovered cost to control a safety-critical system, while the second is emphasizing the cost in itself (supposedly as opposed to being just instrumental for synthesizing an apprentice policy). Could you clarify?


Here are a few minor typos (or subjective suggestions) I noticed:
- "infinitely-uncountably-many constraints" - I think saying "uncountably many" implies infinitely many and thus is sufficient as a shorter term. If you insist on including both, I'd write uncountably-infinitely many, since the uncountably is qualifying the "infinitely".
- l. 108 missing comma after citation
- "\gamma-discount \nu_0 optimal policy" - the "discount" sounds strange. Maybe \gamma-discounting would sound better?
l. 330 "Theorem" -> theorem
l.379 missing closing brackets
l. 550 extra comma before "under"
l. 838 sate -> state
l. 841 performs -> performance

**Limitations:**

- As said, the bounds provided are not always useful in practice yielding completely impractical requirements.
- The applicability to real-world problems is unexplored in the paper and requires further work.

---

> ### Author Rebuttal · Authors · 2024-08-06
>
> We thank the reviewer for the time spent in evaluating our work. In the following, we address your comments.
> > The paper certainly contains enough material for a substantially longer journal paper - some of the sections could benefit from being longer, trying to convey more intuitions and context about the presented theory.
>
> We thank the reviewer for recognizing the substance and novelty in our paper, which could indeed justify an even longer manuscript. We are delighted by your appreciation of our writing style, as we put significant effort into presenting the material in an accessible and comprehensible manner without compromising on mathematical clarity and rigor.
> Given the constraints of the conference paper format and the substantial amount of theoretical concepts we cover, we aimed to present our findings concisely while maintaining coherence. For the final version of the paper, we will utilize the additional page allowed to further elaborate on certain sections and provide additional intuition and context about the presented theory.
>
> > More thorough empirical evaluation would also be beneficial...
>
> We illustrate our method with a simple truncated Linear Quadratic Regulator (LQR) example (see App. C) to provide better intuition about the method and the proposed sample complexity bounds.
> Although the focus of this work is theoretical, we will endeavour to provide a more complex numerical result, possibly in the Supplementary Material. It is worth noting that the work of Dexter et al. [50], the only theoretical study addressing IRL in continuous state (but discrete action) spaces, also includes simulations for a representative one-dimensional state space MDP.
>
>
> > The bounds provided by the theorems don't always seem meaningful ...
>
> Theorem 2.1 provides explicit sample complexity bounds for achieving a desired approximation accuracy with our proposed randomized algorithm. The corresponding sample complexities include the expert sample complexity, the number of calls to the generator, and the number of sampled constraints. The first two complexities scale gracefully with respect to the problem parameters, whereas the number of sampled constraints scales exponentially with the dimension of the state and action spaces. This makes the algorithm particularly suitable for low-dimensional problems of practical interest, e.g., pendulum swing-up control, vehicle cruise control, and quadcopter stabilization. We will include these examples of manageable dimensionality and related references in Rem. 4.1. Moreover, in the uploaded supplementary PDF, we show additional instances of figure (c) for different parameter choices, illustrating the significant effect on the resulting sample complexities.
>
> Note that  a similar exponential dependence to the dimension of the state space has been established in Dexter et al. [50], the only theoretical work on IRL in continuous states and discrete action spaces.
>
> We intend to provide a more detailed discussion on enhancing sample complexity bounds through the utilization of the underlying problem structure in the paper. In addition, it becomes imperative to gain an understanding regarding the selection of a suitable distribution for drawing samples in the future. Intuitively, it is reasonable to anticipate that certain regions within the state-action space carry more "informative" characteristics than others. One conjecture is that sampling constraints based on the expert occupancy measure could yield a more scalable bound. However, a comprehensive mathematical treatment of these inquiries will be addressed in future research endeavors.
>
> > Isn't the identifiability issue a deeper issue than a mere "mathematical artifact"?
>
> The normalization constraint in our formulation primarily aims to address the ill-posedness (or, equivalently, the ambiguity issue) inherent in the IRL problem.
> In particular we state and prove that the normalization constraint rules out a wide class of trivial solutions, i.e., all constant functions and inverse solutions of the form $c=T_\gamma^*u$, an outcome devoid of physical meaning and a mathematical artifact. While the identifiability problem and the ill-posedness problem are related in IRL, they are not the same. Identifiability deals with the uniqueness of the true cost function, whereas ill-posedness is a broader concept from mathematical and statistical problems and refers to situations where a problem does not satisfy the conditions for being well-posed (e.g., in our case due to infinitely many solutions).  Overall, the purpose of the normalization constraint is to exclude a large class of meaningless solutions, rather than to address the identifiability problem. Note that, unlike recent theoretical IRL works [48-53] which avoid discussing this issue altogether, we attempt to address the ambiguity problem and provide theoretical results in this direction, hoping to lay the foundations for overcoming current limitations.
>
> > Do you have an intuitive explanation for why the constraint on the epsilon constraint on the consistency of the cost and value function is enforced point-wise from below and in expectation from above?
>
> The characterization of the inverse feasibility set is due to linear duality and complementary slackness conditions (lines 210-211). In particular, the constraint that holds pointwise is due to dual feasibility and  and the constraint that holds in expectation is due to strong duality (see proof of Thm. 3.1 lines 611-612). The formulation (IP) is a relaxation of the constraints in the inverse feasibility set and paying a
> penalty when violated (lines 288-289). We will add this insight to the revised text.
>
> > Minor points suggestions
>
> We will follow your suggestions and remove the third motivation as well as the indicated sentence about the recovery of the cost function to avoid confusion.
>
> > Minor typos
>
> We thank the reviewer for spoting these typos; We fixed all of them.

---

> ### Comment · Reviewer_p8oS · 2024-08-12
>
> I'd like to thank the authors for a thorough response to the points raised in the review. I'll gladly keep the "accept" recommendation.
>
> Regarding the discussion on
> > Isn't the identifiability issue a deeper issue than a mere "mathematical artifact"?
>
> Just to clarify: I'm happy with your technical solution, and the point I was raising was more of a subtle phrasing issue. What I was objecting to was mainly that in the first paragraph, you mention reward shaping and then you say that "*all* these examples illustrate ... mathematical artifacts". I'd say your solution correctly filters out mathematical artifacts and meaningless rewards. However, the issue of underdeterminacy (e.g. with respect to reward shaping) can remain (unless we have access to e.g. multiple environments) and is important in IRL, rather than being mere mathematical artifact (which is a label I'm ok with in relation to the rewards you're filtering out). Maybe a slight rephrasing may help not to sound like putting all these transformations (incl. general reward shaping) in a single basket.

---

> ### Author Response · Authors · 2024-08-14
>
> Thank you for your valuable feedback. We understand your concern regarding the phrasing in the paragraph, particularly the mention of reward shaping and the reference to "mathematical artifacts.". We will revise the phrasing to ensure that reward shaping and underdeterminacy are not inadvertently grouped with mathematical artifacts, thereby maintaining a clear distinction between them. Indeed, these issues cannot be fully resolved unless, for example, one has access to multiple expert policies or environments for comparison. Thank you again for pointing this out.

---

### Author Rebuttal · Authors · 2024-08-07

We thank all reviewers for their helpful feedback. Below, we provide our responses addressing each point raised by the reviewers.

---

### Decision · Program_Chairs · 2024-09-25

**Decision:**

Accept (poster)

**Comment:**

The reviewers agree that the paper makes significant contributions to the IRL community. Importantly, this is the first work to solve the (reward learning) inverse problem in tabular MDPs with a tractable algorithm. However, in the continuous state-action case, the exponential dependence on $\text{dim}(X \times A)$ is not supported by any lower bound, which raises questions about its tightness. In general, the paper contains a lot of material that, understandably, is difficult to convey within the 8 pages allowed for a conference submission. This makes the paper's organization, in the reviewers' opinion, confusing, as the current version dedicates too much space to the continuous case, whose results (as discussed above) are not entirely satisfactory.

All things considered, I recommend accepting this paper in light of its clear advancement of the IRL literature. **At the same time, I strongly recommend that the authors revise the paper for the camera-ready version, taking advantage of the additional page, to incorporate the reviewers' inputs about the issues with the presentation, e.g., give more space to the results for the tabular case (tractability) and clarify the limitations of the analysis for the continuous case (exponential dependence on $\text{dim}(X \times A)$).**